



# Natural Surface Emissions Dominate Anthropogenic Emissions Contributions to Total Gaseous Mercury (TGM) at Canadian Rural Sites

Irene Cheng[1], Amanda Cole[1], Leiming Zhang[1], Alexandra Steffen[2]

[1]Measurements and Analysis Research Section, Air Quality Research Division, Science and Technology Branch, Environment and Climate Change Canada, Toronto, M3H 5T4, Canada
[2]Processes Research Section, Air Quality Research Division, Science and Technology Branch, Environment and Climate Change Canada, Toronto, M3H 5T4, Canada

*Correspondence to*: Irene Cheng (irene.cheng@ec.gc.ca)

**Abstract.** The Canadian Air and Precipitation Monitoring Network (CAPMoN) measures total gaseous mercury (TGM) at three rural-remote sites. Long-term TGM, ancillary measurements and the Positive Matrix Factorization (PMF) model were

used to assess temporal changes in anthropogenic and natural surface emission (wildfires plus re-emitted Hg) contributions to TGM and examine the emission drivers of the observed TGM trends between 2005 and 2018. TGM showed decreasing trends at the three sites; the magnitudes (ng m$^{-3}$ yr$^{-1}$) were -0.050 at Saturna for 2010-2015, -0.026 at Egbert for 2005-2018, and -0.014 at Kejimkujik for 2005-2016. The increasing contributions from natural surface Hg emissions at Saturna (1.64% yr$^{-1}$) and Kejimkujik (1.03% yr$^{-1}$) resulted from declining anthropogenic Hg emissions and increasing oceanic and terrestrial Hg re-

emissions. The mean relative contributions of natural surface emissions to annual TGM were 65%, 72.5% and 65% at Saturna, Egbert and Kejimkujik. TGM at Saturna were mainly from background Hg (53%), Hg re-emissions (14%), and shipping (10%); at Egbert, from background Hg (63%), Hg re-emissions (15%), and crustal/soil dust (9%); and at Kejimkujik, from background Hg (71%), regional point source emissions (10%), and Hg re-emissions (8%). Local combustion sources contributed a few percent of the annual TGM, while the percentage from oceanic Hg evasion was 6.6-9.5% for the two coastal sites. Wildfire

impacts on annual TGM were 5.6% at Saturna, 1.3% at Egbert, and 2.1% at Kejimkujik. Background Hg contributions to TGM were greater in the cold season, whereas wildfire and surface re-emission contributions can be significant in the warm season.

## 1 Introduction

Mercury (Hg) is a global pollutant that is toxic to biota and human health. Atmospheric Hg comprises three dominant inorganic fractions including gaseous elemental Hg (GEM), gaseous oxidized Hg or reactive gaseous Hg (GOM or RGM) and particle-

bound Hg (PBM). GEM is the predominant form capable of long-range transport and chemical transformation to divalent Hg(II) compounds, such as GOM and PBM. Hg(II) has an atmospheric residence time of several weeks and therefore tends to deposit locally or regionally. Through dry and wet deposition, atmospheric Hg enters terrestrial and aquatic environments,



where it undergoes conversion to methylmercury (MeHg) compounds. The bioaccumulation of MeHg is known to cause severe neurological and reproductive effects (Hong et al., 2012; Driscoll et al., 2013; Obrist et al., 2018).

Emissions drive atmospheric Hg transport, chemical processing, deposition, and subsequent ecosystem impacts. Hg sources to the atmosphere consists of natural emissions (e.g., biomass burning, volcanic and geothermal releases), anthropogenic emissions (e.g., coal combustion, metal smelting, cement production), and re-emissions from terrestrial and aquatic surfaces. Globally, primary anthropogenic Hg emissions account for approximately 30% of the total atmospheric Hg emissions with the remainder from natural emissions and surface re-emissions. The latest global inventories indicate anthropogenic sources

emitted 2200-2500 Mg Hg yr$^{-1}$ (Dastoor et al., 2024 and references therein). 200-500 Mg yr$^{-1}$ are emitted from geothermal and volcanic releases and 300-680 Mg yr$^{-1}$ from wildfires. Terrestrial and aquatic surface re-emissions account for 1000-1700 Mg yr$^{-1}$ and 2800-8300 Mg yr$^{-1}$, respectively (Pirrone et al., 2010; Outridge et al., 2018; Streets et al., 2019; Li et al., 2020; Shah et al., 2021; Feinberg et al., 2022; Sonke et al., 2023). After accounting for atmospheric deposition, the global atmospheric Hg budget is in the range of 3800-4500 Mg. Natural emissions and surface re-emissions of Hg have large uncertainties because of

limited spatial and temporal measurements for constraining emission factors. There are also uncertainties with Hg air-surface exchange models and quantifying the proportion of re-emitted Hg originating from natural and anthropogenic sources (Dastoor et al., 2024).

Source attribution analyses, which study the linkages between emission sources and environmental concentrations, have been conducted using receptor methods and chemical transport models (CTMs). Receptor methods, such as Positive Matrix

Factorization (PMF) model, principal components analysis (PCA) and back trajectory analysis, are applied to ambient air measurements at a particular location (Cheng et al., 2015 and references therein), whereas CTMs integrate detailed information on Hg emissions, meteorology, chemistry and deposition to predict ambient Hg concentrations typically over a broader region (Horowitz et al., 2017; Shah et al., 2021; Zhang et al., 2023). Receptor methods have been used to identify sources contributing to GEM, GOM, PBM, total gaseous mercury (TGM) (Mazur et al., 2009; Huang et al., 2010; Eckley et al., 2013; Wang et al.,

2013; Cheng et al., 2017; Qin et al., 2020; Custodio et al., 2020), and Hg wet deposition (Keeler et al., 2006; Michael et al., 2016). There are numerous modeling studies on atmospheric Hg with some focusing on an improved Hg emissions budget (Fisher et al., 2012; Zhang et al., 2016, 2023), chemical mechanisms (Shah et al., 2016), vegetation uptake (Zhou et al., 2021), and source apportionment estimates (ECCC, 2016; Fraser et al., 2018; Dastoor et al., 2021, 2022).

Anthropogenic Hg emission sources inferred from receptor methods include coal, wood and oil combustion, cement

production, iron and steel production, and vehicular traffic. Receptor methods have also been used to assess the effectiveness of emissions control on ambient Hg concentrations. PMF modeling showed that the closure of coal-fired power plants resulted in significant reductions in Hg source contributions to a nearby monitoring site (Wang et al., 2013). Concentration-weighted trajectory analysis revealed steep declines in Hg source contributions over southeastern Canada and northeastern U.S., which were driven by Hg emission reductions from power plants (Cheng et al., 2017). PCA also showed the diminished point source

impact on TGM following the closure of a local base metal smelter in Canada (Eckley et al., 2013). Aside from anthropogenic





Hg sources, source attribution studies have revealed the growing importance of natural Hg emissions (e.g., wildfires, oceanic evasion), chemical processing, and long-range transport.

The objective of this study is to determine the source contributions to TGM at three Canadian Air and Precipitation Monitoring Network (CAPMoN) sites using the Positive Matrix Factorization (PMF) model and assess their long-term changes. Sources
identified from the PMF model are aggregated into anthropogenic and natural surface emission contributions to examine the changes in their relative proportions over time. The long-term CAPMoN TGM measurements will also be used to conduct statistical trends analysis and assess the emission drivers of the long-term trends.

## 2 Methods

### 2.1 TGM measurements and ancillary data

The CAPMoN TGM sites are located in Saturna, British Columbia; Egbert, Ontario; and Kejimkujik National Park, Nova Scotia. Saturna (SAT) is a coastal site in western Canada close to the Pacific Ocean. Egbert (EGB) is an inland site in southeastern Canada. Kejimkujik National Park (KEJ) is a coastal site in eastern Canada close to the Atlantic Ocean. The three sites are in rural-remote locations with minimal influence from large emission sources; thus, TGM concentrations at the sites are regionally representative. Refer to Table 1 for additional site information. TGM at other Canadian sites has been analyzed
in previous publications (Temme et al., 2007; Cole et al., 2014; MacSween et al., 2022). The CAPMoN TGM sites were selected for this study to provide an update on current concentrations and patterns. Ancillary measurements are also available at the sites to conduct PMF analysis.

TGM is measured using a Model 2537 Hg vapor analyzer (Tekran Instruments Corporation, Toronto, Ontario, Canada), which employs cold vapor atomic fluorescence spectrometry (CVAFS) to quantify TGM. The analyzer is housed in a shelter at each
location with the sampling line extending through the rooftop to the outdoor environment. The approximate height of the air inlet above ground is 6 m. Ambient air is sampled at 1.5 L min$^{-1}$ through a heated PFA line maintained at temperatures at least 10°C above ambient. A Teflon filter at the air inlet and back of the analyzer removes particulate matter from the air stream. TGM, including GEM and GOM, is adsorbed onto dual gold cartridges which alternate between sampling and desorption every 5 min. TGM concentrations are reported in units of ng m$^{-3}$ referenced to STP (0°C, 1 atm). Automated calibrations are
performed every 25 h using the analyzer's Hg permeation source, verified annually by manual injections of a known amount of Hg$^0$ to zero air using the Tekran Model 2505 Calibration Unit. The permeation rate derived from manual calibrations is expected to be ± 5% of the analyzer's perm rate. Every 35 samples, the permeation source releases Hg$^0$ into the sampled air to assess recovery.

Site operators and field staff perform instrument checks and maintenance activities including sample filter replacement, lamp
voltage adjustments, argon tank replacement, etc. Site audits are conducted annually, including verification of the Hg permeation rate and sample flowrate, leak and sample line checks, and gold cartridge replacement if necessary. The 5-min TGM data are quality controlled by flagging instrument parameters (baseline voltage mean and standard deviation, cartridge





difference, sample volume, etc.) that are outside the normal range of operations. The data reviewer checks field notes, site audit reports and wildfire maps, verifies quality control flags, investigates anomalies and outliers, and computes hourly

averages for final reporting.

Ancillary measurements of particulate inorganic ions, $SO_2$, CO, total carbon, and air temperature were obtained from CAPMoN and other monitoring networks (Table S1). Inorganic ions, $SO_2$, and temperature were measured at the sites, whereas regionally representative CO and total carbon measurements were within 40 km of the sites. All data have been quality controlled by their respective networks and are publicly accessible. Hourly data were converted to 24-h averages for input into the PMF

model.

## 2.2 PMF model

The USEPA PMF model is a multivariate model for the source apportionment of air pollutants, such as speciated particulate matter, VOC, trace elements and speciated atmospheric Hg. The principles behind the PMF model are detailed in previous studies and the user's guide (Norris et al., 2014; Brown et al., 2015). The model input for this work included a dataset consisting

of 24-h chemical species concentrations and mean air temperature (Table S1). Uncertainties were calculated using the equation provided in the user's guide, which is a function of the detection limit (DL) and error fraction (Norris et al., 2014). Based on pollutant tracers, factor profiles were assigned to mercury sources using known source profiles or emissions information from literature. The main assumption of the PMF model is that no chemical transformation takes place during atmospheric transport. Thus, we limited the input of reactive or secondary species into PMF. The input of TGM is advantageous over speciated Hg

because TGM is the sum of GEM and GOM (oxidized Hg) which tends to be less impacted by chemical reactions.

PMF model runs were performed for each year of data separately resulting in yearly factor profiles. We ran the model using 6 factors and attempted additional runs with 5 and 7 factors. The regression fit between modeled and observed TGM and interpretability of the factors among the yearly runs were assessed to determine the optimal number of factors. The model-observed $R^2$ values for TGM were $\geq 0.5$ for all yearly runs. Plots of PMF modeled versus observed 24-h TGM are shown in

Fig. S1. The plots illustrate strong correlations between modeled and observed TGM at the three sites with $R^2$ of 0.71-0.76. The modeled TGM also reproduced the time-series of the observed TGM quite well except for a few elevated TGM concentrations at KEJ in early 2005, which suggests that the final 6-factor PMF solution was a good fit to the observations.

There was one factor in the PMF solution that could represent both surface re-emissions and wildfires. To distinguish between surface re-emission and wildfire contributions, we examined the covariance in the source contributions and fire radiative power

(FRP) from MODIS data (NASA, 2023). We identified a FRP threshold for screening wildfire-influenced daily source contributions. Source contributions below the FRP threshold were assumed to be from surface re-emissions.

Anthropogenic and natural surface emission contributions were derived from source contributions. Factors representing local or regional combustion and secondary sulfate were classified as anthropogenic. Natural surface emissions include factors representing wildfires, terrestrial surface re-emissions (GEM re-emissions, dust resuspension), and oceanic evasion. The

definition of natural surface emissions discussed in this paper does not necessarily refer to the natural origin of the emissions.



Hg re-emissions from land and water surfaces originate from both natural and anthropogenic sources; however, their relative contributions remain uncertain. Background Hg refers to the northern hemispheric Hg pool, which is also attributed to both natural surface and anthropogenic Hg emissions. We assumed the natural surface/anthropogenic contribution was proportional to that of global Hg emissions reported in Pirrone et al. (2010) and Streets et al. (2019), which was estimated to be 135   68.5%/31.5%.

## 2.3 Long-term trends analysis

Long-term trends analyses were performed using the Theil-Sen slope estimator. The analysis was applied to observed daily TGM concentrations and TGM source contributions (absolute and relative). We computed the annual rate of change and those for the cold (Nov to Apr) and warm (May to Oct) seasons separately to assess whether the trends differ. The TGM analysis 140   period is indicated in Table 1; however, some years of data were unavailable for statistical trends analysis. At SAT, the 2016 data did not meet the data completeness threshold (50% of data available for each season) for trends analysis. TGM concentrations were not measured in 2017 at SAT and thus, the trends analysis was not extended to 2018. KEJ was relocated 3 km south of the original site in February 2017, and the TGM analyzer was changed from the Tekran 2537B to 2537X model. Concurrent TGM measurements at KEJ and the new site (KEB) indicate TGM was significantly higher at KEB. The monthly 145   mean hourly TGM absolute and relative differences were 0.2-0.29 ng m$^{-3}$ and 16.6-21.5%. The cause of the difference may have been the change in instrument model but this is inconclusive and therefore, the 2017-2018 data at the new site were excluded from the long-term trends analysis.

**Table 1: CAPMoN TGM site information**

| Site ID (site name, province) | Latitude, longitude | Time zone (UTC offset) | Site characteristics | Period of data analyzed |
|---|---|---|---|---|
| SAT (Saturna, British Columbia) | 48.775, -123.128 | PST (UTC-8) | Coastal site near the Pacific Ocean | 2009-2018 (no data collected in 2017) |
| EGB (Egbert, Ontario) | 44.231, -79.783 | EST (UTC-5) | Inland site in southeastern Canada, agricultural activities nearby | 2005-2018 |
| KEJ (Kejimkujik National Park, Nova Scotia) | 44.432, -65.203 | AST (UTC-4) | Coastal site near the Atlantic Ocean, forested area nearby | 2005-2016 |




## 3 Results

### 3.1 Saturna

#### 3.1.1 Overview of TGM concentrations

The range in annual mean TGM concentrations were 1.14-1.49 ng m$^{-3}$ at Saturna (SAT) during 2009-2018. Annual descriptive
statistics are summarized in Supplement Table S2. The variability in 24-h TGM was assessed using relative standard deviation
(RSD). The RSD was 9.7% and decreased over time. The 75$^{th}$ percentile concentrations varied between 1.22 and 1.61 ng m$^{-3}$
depending on the year (Fig. 1). Mean TGM was highest in the spring (Mar - May) and lowest in the fall (Sep - Nov); the means
were comparable for the winter (Dec - Feb) and summer (Jun – Aug). The seasonal pattern at SAT was consistent with those
reported for other Canadian and northern hemisphere rural-remote sites (Cole et al., 2014; ECCC, 2016). Diel patterns in TGM
differed between seasons. The diel amplitude was 4% for winter, 6% for spring, 14% for summer, and 4% for fall. In addition
to the strong diel cycle in the summer, TGM also peaked earlier (7:00-8:00 PST) than in other seasons. Spring and fall peaks
in hourly TGM were broader typically occurring during 8:00-11:00 PST, whereas the hourly TGM peak occurred during 12:00-
13:00 LST in the winter. Lower TGM was typically observed during the evening and nighttime. During the 2009-2018 period,
the highest and lowest annual TGM were observed in 2010 (1.49 ng m$^{-3}$) and 2015 (1.14 ng m$^{-3}$), respectively (Fig. 1). Annual
TGM decreased from 2009 to 2015 and then returned to higher concentrations in 2016 and 2018.

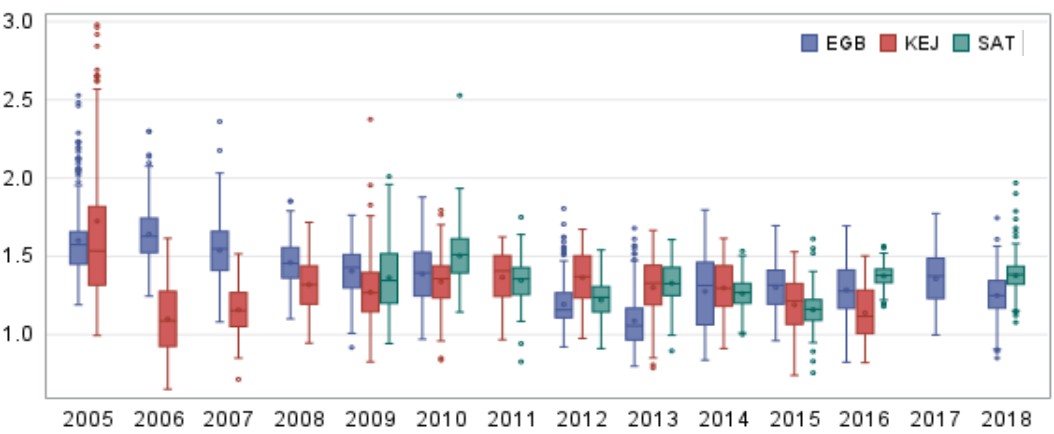


**Figure 1: Box-whisker plots of 24-h average TGM concentrations at Egbert (EGB), Kejimkujik National Park (KEJ) and Saturna (SAT)**

#### 3.1.2 PMF factor profiles

PMF model runs were undertaken separately for each year. For SAT, six factors were generated for each year. Based on the
analysis of key variables, the factors represented aged sea-salt aerosols (SSA), fresh SSA, local combustion, background Hg,





secondary sulfate, and terrestrial GEM re-emissions/biomass burning. The mean percentage of the total factors investigated are shown in Fig. 2 and reported below.

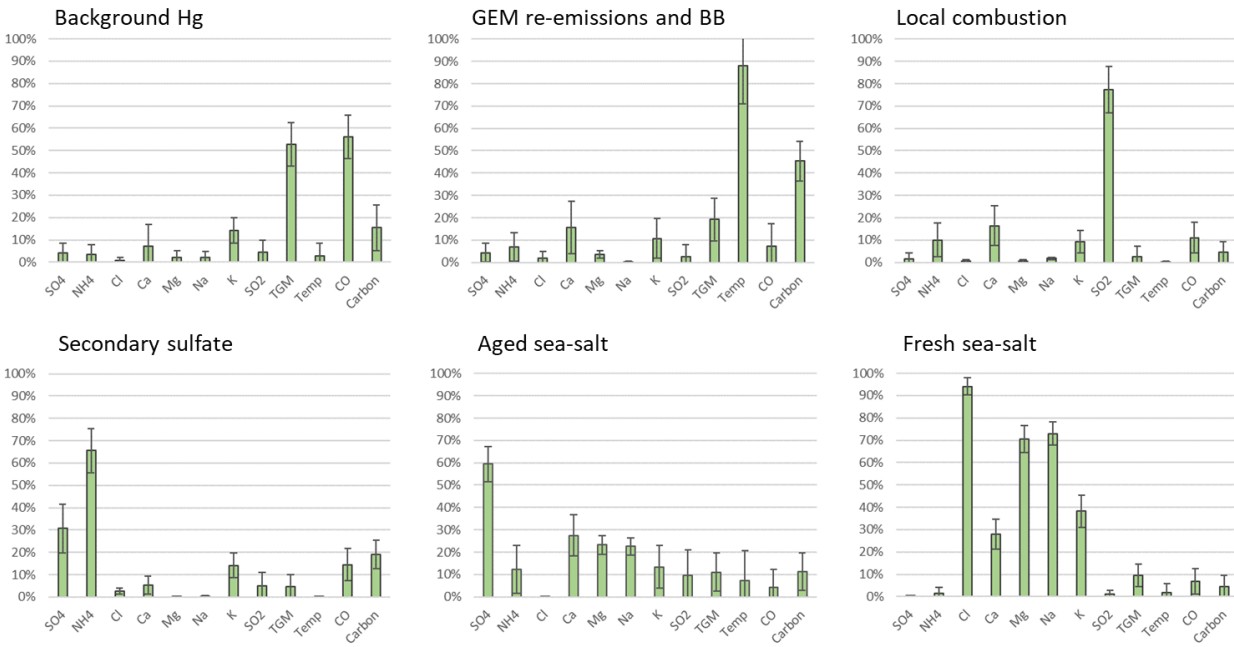

**Figure 2: PMF mean factor profiles for SAT. Error bars indicate standard deviation among the years; BB: biomass burning.**


The aged SSA factor was characterized by moderate abundance of $Ca^{2+}$, $Mg^{2+}$ and $Na^+$, elevated $SO_4^{2-}$ and negligible $Cl^-$ (Fig. 2), which is a result of the chemical reaction between fresh SSA and acidic compounds. In contrast, the fresh SSA factor had high abundance of $Na^+$ and $Cl^-$ and negligible $SO_4^{2-}$. The TGM percentage in the aged and fresh SSA factors were ~10% each. The combustion factor was characterized by high abundance of $SO_2$ (77%). This could include local fossil fuel combustion

and metal smelting, which are both significant sources of $SO_2$ and Hg. The TGM percentage in the combustion factor was only 3%. Background Hg was identified based on the high abundance of CO (56%). The TGM abundance was the highest in this factor (53%). Background Hg refers to the northern hemispheric Hg pool, which consists of natural, anthropogenic, and re-emitted Hg that is subject to long range transport (Selin et al., 2008). In previous studies, Hg/CO ratios were used to infer long range transport from Asia to western North America (Weiss-Penzias et al., 2007); hence, CO was used as a tracer of background

Hg. The secondary sulfate factor had moderate abundance of $SO_4^{2-}$ and high abundance of $NH_4^+$; these chemical species were likely produced via the reaction between gaseous ammonia and acidic gases. The TGM percentage in the sulfate factor was 5%. The last factor was representative of GEM re-emissions and/or biomass burning, identified by the relatively higher temperature in this factor (88%) and presence of total carbon (45%), and $K^+$ (11%). Warm temperatures are conducive to the production of wildfires and GEM volatilization from land and water surfaces. Total carbon and $K^+$ are also typically emitted

from wildfires. The TGM percentage in this factor was 10%.



### 3.1.3 Overview of TGM source contributions

The mean relative contribution to annual TGM for the 2010-2018 period at SAT was 65.0% from natural surfaces with the remainder from anthropogenic emissions. This corresponds to mean annual TGM contributions of 0.85 ng m$^{-3}$ and 0.46 ng m$^{-3}$, respectively. The breakdown of TGM source contributions (concentrations and percentage basis) are shown in Fig. 3. TGM
was apportioned to background Hg (52.9%), terrestrial GEM re-emissions (13.7%), shipping emissions and SSA processing (10%), and oceanic evasion (9.5%). Wildfires (5.6%), secondary sulfate (4.6%), and local combustion (3.8%) contributed a smaller fraction of the annual TGM. Shipping emissions in the Ports of Vancouver and Victoria and along the Strait of Georgia had been a significant source of $SO_2$ and $PM_{2.5}$ because of the high sulfur content in marine fuels prior to regulations taking into effect in 2012 (Anastasopolos et al., 2021). Marine fuels also contain heavy metals, such as V, Ni and Hg. As $SO_2$ is
transported downwind and undergoes chemical transformation, acidic gases react with fresh SSA in the MBL resulting in the formation of aged or processed SSA. The assignment of shipping emissions to SSA processing, instead of local combustion, is discussed further in section 3.1.4. Oceanic evasion of GEM was inferred from the fresh SSA factor. Oceanic evasion accounts for more than half of the natural surface Hg emissions globally (Pirrone et al., 2010). SSA also readily adsorb GOM (Rutter and Schauer, 2007a,b; Malcolm et al., 2009), with an estimated 80-90% of the $Hg^{2+}$ bound to SSA (Holmes et al., 2009).
Although the TGM analyzer does not measure PBM, the partitioning of GOM from SSA to the gas phase or $Hg^0$ in the liquid phase partitioning to the gas phase (Malcolm et al., 2003; Rutter et al., 2007a,b; Subir et al., 2012) can also contribute to TGM. There is some uncertainty on the degree of Hg partitioning because as noted in the aforementioned studies Hg gas-particle partitioning is dependent on temperature and particulate matter speciation.

The average relative contribution of anthropogenic TGM was ~35% each for the warm and cold season (Fig. S2), indicating
minimal seasonal differences. Figure S2 illustrates background Hg contributed a larger TGM percentage in the cold season than warm season (65 % vs. 44 %). In contrast, GEM re-emissions, wildfires, and shipping emissions and SSA processing contributed larger TGM fractions in the warm season than cold season. The TGM wildfire contribution for the warm season was six fold greater than that of the cold season. For terrestrial GEM re-emissions, the TGM contribution in the warm season was double that of the cold season. On average, relative contributions from local combustion and secondary sulfate were
comparable between seasons.

### 3.1.4 Interannual and daily variability

Interannual variability in the relative contribution of anthropogenic TGM was in the range of 28-48% (Fig. 3). Anthropogenic contributions were highest in the earlier period (2010 and 2012) because of the larger contributions from shipping emissions and SSA processing. In 2013, 2014 and 2018, natural surface contributions dominated as a result of greater TGM contributions
from terrestrial and marine re-emissions. There were also enhanced TGM contributions from wildfires in 2018 (14%), which was corroborated by observed wildfire statistics. In 2018, the total number of fires and area burned in British Columbia were 39% and 3 times, respectively, above the 2010-2018 average (B.C., 2024). Wildfire characteristics such as location, area



burned, and type of biomass burned, can change year to year and lead to variability in the wildfire Hg emissions. The impact of wildfires and oceanic evasion on SAT also depend on transport patterns and meteorology, which vary interannually.


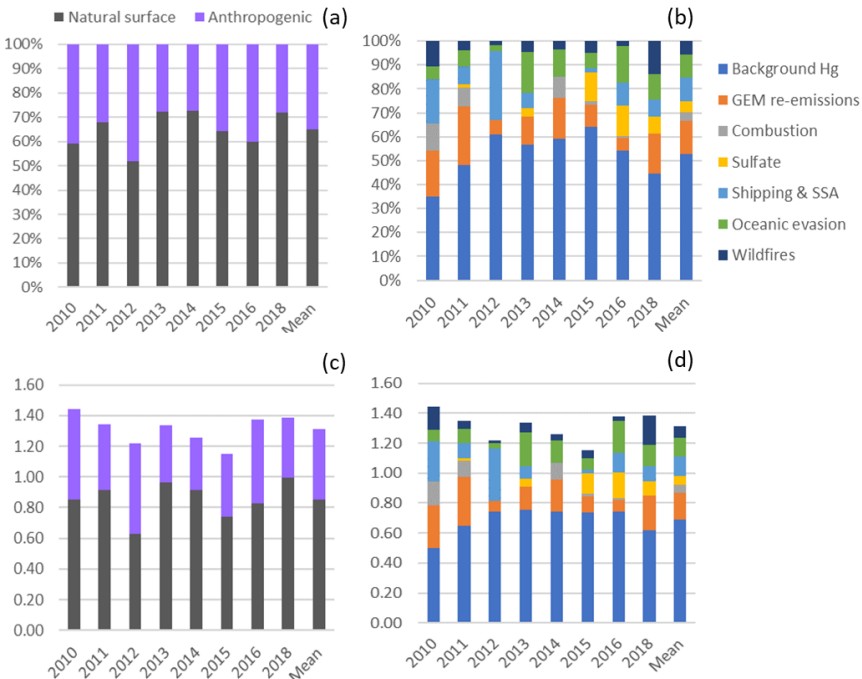

**Figure 3: Impact of natural surface emissions, anthropogenic emissions, and individual emission sources on annual TGM at Saturna (SAT). (a) and (b) plots are relative contributions; (c) and (d) are contributions expressed in concentrations (ng m⁻³). Mean applies to the 2010-2018 period except 2017. Natural surface emissions comprise wildfires, GEM re-emissions, oceanic evasion, and natural**
**surface emissions contributing to background Hg. Anthropogenic emissions comprise local combustion, sulfate, shipping and sea-salt processing, and anthropogenic emissions contributing to background Hg.**

Figure 4 shows the variation in the daily percentage TGM contributions from natural surface and anthropogenic emissions for SAT and other sites. Relative contributions from natural surface emissions dominated anthropogenic emissions except on a
few occasions. Natural surface emissions contributed 60-100% to daily TGM on most days and were greater during summertime.





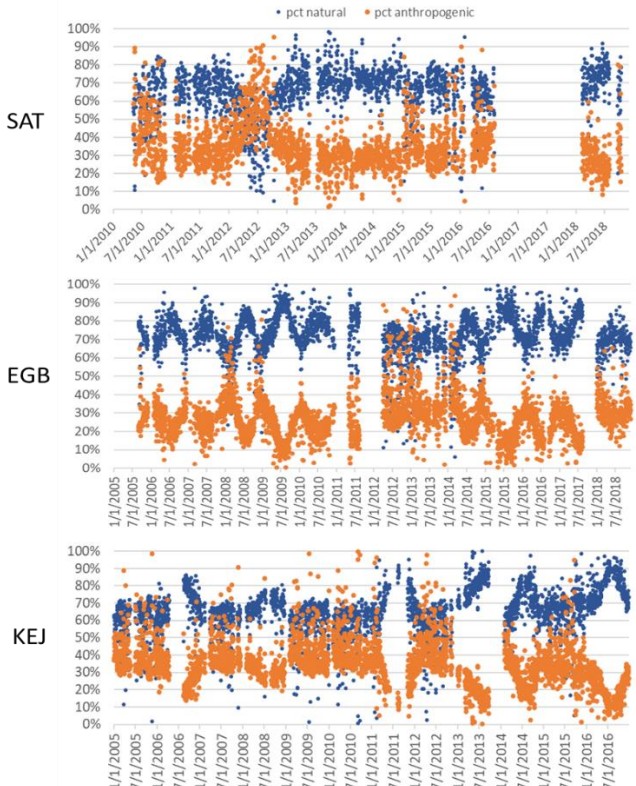

**Figure 4: Relative contributions of natural surface (wildfires plus re-emissions) and anthropogenic emissions to daily TGM at SAT, EGB and KEJ.**


The corresponding plot for variation in individual source contributions at SAT is shown in Fig. 5. There were pronounced seasonal patterns for background Hg, terrestrial GEM flux, and wildfire emission contributions. Relative TGM contributions from background Hg exhibited a consistent seasonal pattern year over year. Daily relative TGM contributions from background Hg can exceed 90% during winter and fall below 30% during summer (Fig. 5). In contrast, those from GEM re-emissions and

wildfires were more variable year over year with lower relative TGM contributions in the summer of 2012 and 2016 and higher contributions in the summer of 2010, 2011 and 2018. Daily TGM contributions from background Hg exceeded 1 ng m$^{-3}$ on a regular basis across the time-series. Those from GEM re-emissions and wildfires were rarely above 1 ng m$^{-3}$.

Relative TGM contributions from local combustion, secondary sulfate, and shipping emissions and SSA processing were episodic. There were a few years with relative daily contributions above 30% and a few years without any contributions.





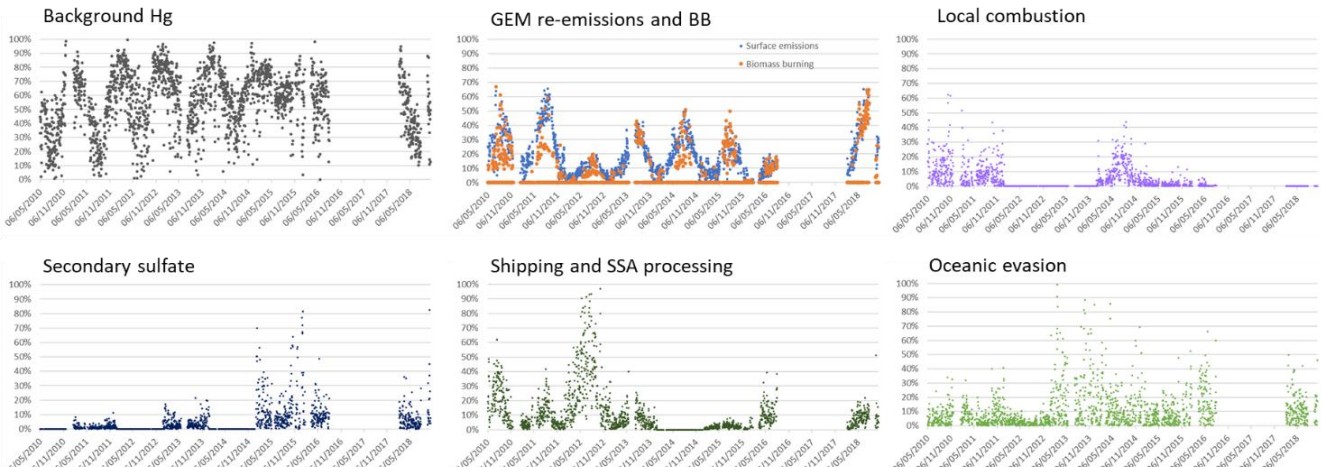


**Figure 5: Relative contributions to 24-h mean TGM from various sources at SAT**

Shipping emissions and SSA processing contributions were noticeably greater than that of oceanic evasion for the earlier period, whereas the case was reversed after 2013. The stronger signature in the earlier period was likely due to higher $SO_2$

concentrations from shipping, resulting in enhancements of acidic compounds in the air to react with fresh SSA. After regulations to limit sulfur content in marine fuels were implemented in 2012, there were substantial decreases in both $SO_2$ and $PM_{2.5}$ concentrations near Canadian port cities (Anastasopolos et al., 2021). According to the Canadian emissions inventory, in addition to $SO_2$ emissions, Hg emissions from marine transportation also decreased from 2010 to 2015 (Fig. S3). This was likely a co-benefit of reducing sulfur in marine fuels. $SO_2$ and Hg emissions in 2015 were 3.6% and 5.4%, respectively, of the

2010 levels. As $SO_2$ emissions from shipping declined, TGM contributions from fresh SSA increased. We infer this source to be oceanic Hg evasion. The downward trends in Hg and $SO_2$ from shipping emissions and aged SSA production differed from the lack of trends in Hg contributions from local combustion (Fig. 5) and their Hg emission trends, e.g. metals and cement production (Fig. S4). Thus, the shipping emission was not combined with the local combustion factor.

### 3.1.5 Long-term trends

Statistical trends analysis was performed on long-term TGM measurements and PMF modeled TGM source contributions, in order to assess potential causes of TGM trends. Both the observed and PMF modeled TGM showed decreasing trends at SAT with similar magnitudes (-0.0497 ng m$^{-3}$ yr$^{-1}$, p<0.05) for the 2010-2015 period (Fig. 6). The comparability in the trends (red line in Fig. 6) and seasonal means (blue line) indicate the PMF model reproduced the TGM observations and adequately captured the seasonal variability over the entire time-series.






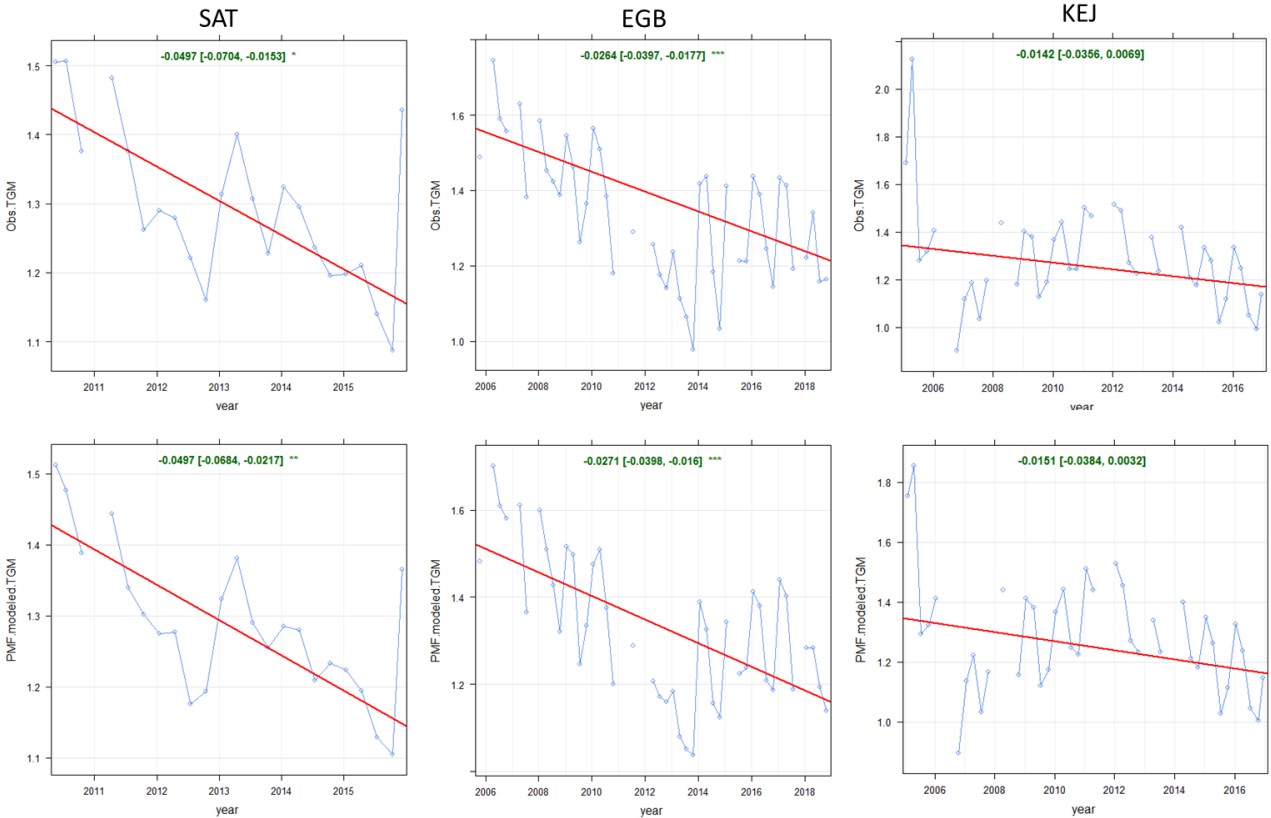

**Figure 6: Long-term trends in observed and PMF modeled TGM concentrations at SAT, EGB and KEJ. Blue line: observed or modeled TGM; red line: trendline; green text: slope of the trendline (ng m$^{-3}$ yr$^{-1}$)**

TGM contributions from background Hg showed a slight increasing trend, but it was not statistically significant and opposite in direction of the trends for observed and modeled TGM concentrations (Table 2). TGM contributions from terrestrial GEM re-emissions (-0.0284 ng m$^{-3}$ yr$^{-1}$, $p<0.1$) and shipping and SSA processing (-0.041 ng m$^{-3}$ yr$^{-1}$, $p<0.001$) decreased significantly with time. No statistically significant trend was also found for TGM contributions from local combustion. Within 150 km of SAT, the major combustion sources of Hg were cement manufacturing and primary metal production (Fig. S4). Hg emissions

from cement production in British Columbia showed an increasing trend from 2010 to 2015, but the same source type showed a decreasing trend in Washington State. Except for 2010, Hg emissions from metal production were stable. Overall, emissions inventory data support the lack of trend in the TGM contributions from local combustion. Secondary sulfate, oceanic Hg evasion, and wildfire TGM contributions did not show statistically significant trends (Table 2). Overall, the reduction in TGM contributions from GEM re-emissions and from shipping and SSA processing were the main drivers of the decreasing TGM

concentrations at SAT.





**Table 2: Long-term trends in TGM concentrations and source contributions from the PMF model. SAT: 2009-2015, EGB: 2005-2018, KEJ: 2005-2016. Statistically significant (p<0.1) trends are shown in bold.**

| | | Trend | | Significance |
|---|---|---|---|---|
| Site | Parameter/Source | Slope (ng m$^{-3}$ yr$^{-1}$) | Slope (% yr$^{-1}$) | p-value |
| SAT | Obs TGM | **-0.0497** | **-3.46** | 0.010 |
| | Modeled TGM | **-0.0497** | **-3.49** | 0.007 |
| | Background Hg | 0.0453 | 7.70 | 0.154 |
| | Shipping and SSA processing | **-0.0410** | **-21.43** | <0.001 |
| | Terrestrial GEM re-emissions | **-0.0284** | **-13.24** | 0.070 |
| | Local combustion | -0.0125 | -17.26 | 0.212 |
| | Secondary sulfate | 0.0126 | na | 0.102 |
| | Oceanic evasion | 0.0094 | 14.10 | 0.114 |
| | Wildfires | -0.0022 | -6.05 | 0.247 |
| EGB | Obs TGM | **-0.0264** | **-1.69** | <0.001 |
| | Modeled TGM | **-0.0271** | **-1.79** | <0.001 |
| | Background Hg | -0.0171 | -1.76 | 0.140 |
| | Road salt | -0.0012 | -5.21 | 0.142 |
| | Terrestrial GEM re-emissions | -0.0072 | -3.24 | 0.210 |
| | Local combustion | 0.0006 | 0.89 | 0.808 |
| | Secondary sulfate | **-9.69E-8** | **-7.18** | 0.028 |
| | Wildfires | -0.0010 | -6.11 | 0.120 |
| | Crustal/soil | 0.0041 | 4.84 | 0.431 |
| KEJ | Obs TGM | -0.0142 | -1.06 | 0.147 |
| | Modeled TGM | -0.0151 | -1.12 | 0.137 |
| | Background Hg | **-0.0217** | **-2.00** | 0.057 |
| | Regional emission and SSA | **-0.0191** | **-7.62** | <0.001 |
| | Terrestrial GEM re-emissions | **0.0039** | **-37.77** | 0.042 |
| | Local combustion | 0 | 0.00 | 0.576 |
| | Sulfate | 0 | na | 0.876 |
| | Oceanic evasion | 0.0035 | 6.64 | 0.200 |
| | Wildfires | 0.0003 | -24.76 | 0.135 |

Figure 7 shows the trends analysis results for the relative source contributions to annual TGM. The percentage contribution from anthropogenic Hg emissions decreased by 1.64 % yr$^{-1}$ (p<0.05) during 2010-2015. The decline was driven by the reduction in the percentage contribution from shipping emissions and SSA processing (-2.92 % yr$^{-1}$, p<0.001). In contrast, relative TGM contributions from background Hg showed an increasing trend of 4.7 % yr$^{-1}$ (p<0.05). The percentage contributions from fresh SSA and secondary sulfate also increased slightly (<1 % yr$^{-1}$, p<0.1).





**Figure 7: Long-term trends in relative source contributions at SAT. (a) natural surface emissions (wildfires plus re-emitted Hg), (b) anthropogenic emissions, (c) background Hg, (d) shipping and SSA processing, (e) oceanic evasion, (f) GEM re-emissions, (g) biomass burning, (h) local combustion, (i) secondary sulfate. Blue line: relative contributions; red line: trendline; green text: slope of the trendline (x100% yr⁻¹).**

### 3.1.6 Cold and warm season trends

Seasonal trends analyses were also performed, and the Theil-Sen's slopes are summarized in Table S3. The declining trend in the warm season (p<0.1) was greater than that of the cold season for observed and PMF modeled TGM. Background TGM contributions shows an increasing trend of 0.043 ng m⁻³ yr⁻¹ in the warm season (p<0.05), whereas no significant trend was





found in the cold season. TGM contributions from shipping and SSA processing showed a significant decreasing trend in the

warm season (p<0.1). No significant trends were found for seasonal TGM contributions from other sources.

Trends in relative source contributions for the cold and warm seasons were also assessed (Table S4). The proportion of TGM from shipping and SSA processing decreased significantly at a rate of 3.7% yr$^{-1}$ (p=0.07) in the warm season, whereas no trend was found in the cold season. There was a significant increasing trend in the relative TGM contributions from background Hg in the warm season (5.5% yr$^{-1}$, p<0.001). The seasonal trends analyses indicate the annual trends in relative TGM source

contributions from background Hg (increasing) and shipping and SSA processing (decreasing) were primarily driven by warm season trends.

### 3.2 Egbert

#### 3.2.1 Overview of TGM concentrations

The range in annual mean TGM concentrations was 1.08-1.64 ng m$^{-3}$ at Egbert (EGB) during 2005-2018 (Fig. 1; Table S1).

The RSD of 24 h mean TGM was 13.3% and varied from year to year. The 75$^{th}$ percentile concentration ranged between 1.17 and 1.75 ng m$^{-3}$ at EGB depending on the year (Fig. 1). The mean TGM was highest in the winter followed by spring, summer, and fall, respectively. The TGM diel amplitude was 4-5% for winter to summer and 7% in the fall. All seasons showed a broad daytime TGM peak, in which the maximum occurred during 11:00-13:00 LST in winter, summer and fall and during 8:00-10:00 LST in spring. Lower TGM was observed in the evening and nighttime during winter and spring, whereas it was observed

at 6:00 LST during summer and fall. For the 2005-2018 period, annual TGM was highest in 2006 and lowest in 2013. There was a sharp decrease in annual TGM from 2005 to 2013 followed by a rebound in 2014 with concentrations remaining stable during 2014-2018 (Fig. 1).

#### 3.2.2 PMF factor profiles

Six factors were selected for EGB in the final PMF solution. The factors were assigned to background Hg, terrestrial GEM re-

emissions/biomass burning, local combustion, secondary sulfate, SSA, and crustal/soil dust (Fig. 8). Background Hg had the highest abundance of TGM (63%) and CO (60%). Terrestrial GEM re-emissions/biomass burning was characterized by the strong presence of temperature (97%), K$^+$ (34%) and total carbon (43%); the TGM abundance was 17%. The proportions of SO$_2$ and TGM in the local combustion factor were 85% and 6%, respectively. Secondary sulfate was dominated by the presence of sulfate (72%) and ammonium (78%) with a small percentage of TGM (1.8%). A factor was assigned to road salt because of

the high abundance of Na$^+$ (79%) and Cl$^-$ (85%), and the relatively greater contributions during the cold season than warm season. The TGM abundance in the road salt factor was 4.1%. Crustal/soil dust was characterized by the high abundance of Ca$^{2+}$ (82%) and Mg$^{2+}$ (77%); the TGM percentage from crustal/soil was 8.7%. Soil dust emissions are common around the EGB site because of agricultural areas nearby. Contributions of Ca$^{2+}$ and Mg$^{2+}$ peaked in May and September, which coincide with periods of increased agricultural activity.





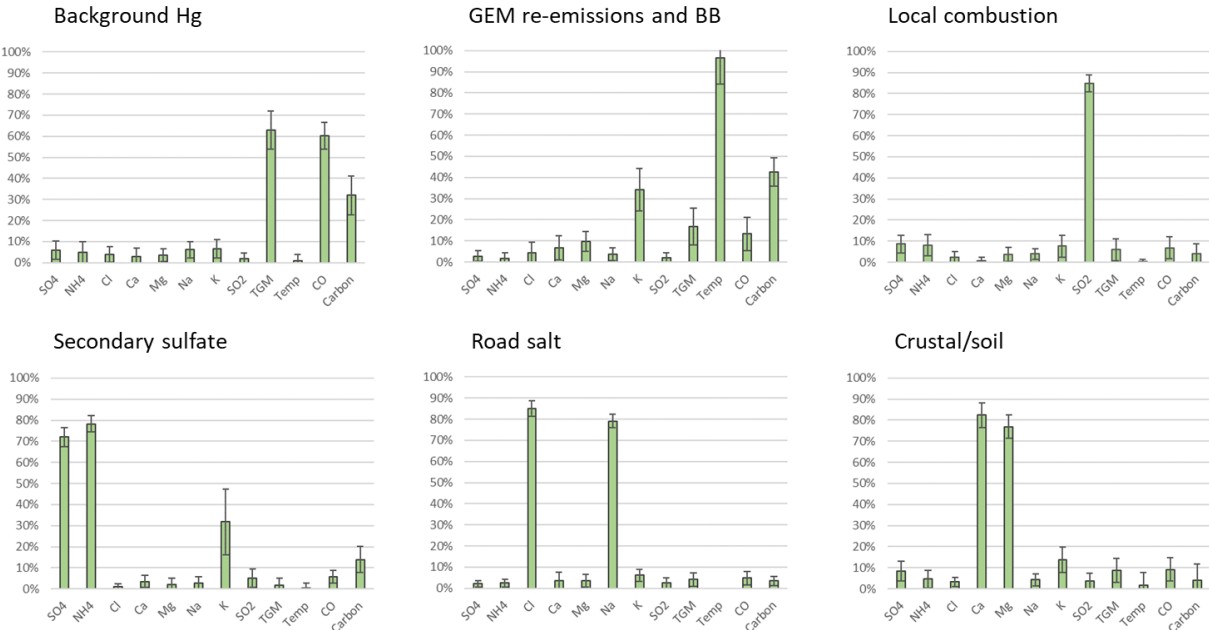

**Figure 8: PMF mean factor profiles for EGB. Error bars indicate standard deviation among the years; BB: biomass burning.**

### 3.2.3 Overview of TGM source contributions

The mean relative contribution of anthropogenic emissions to annual TGM was 27.5% at EGB. Natural surface and anthropogenic emissions contributed 0.98 ng m$^{-3}$ and 0.37 ng m$^{-3}$ to the annual mean TGM concentration. Background Hg (62.9%) contributed the most to annual TGM at EGB followed by terrestrial GEM re-emissions (15.4%), crustal/soil dust (8.7%), local combustion (5.9%), road salt (4.1%), secondary sulfate (1.8%), and wildfires (1.3%), respectively (Fig. 9). Crustal/soil dust and road salt were grouped with GEM re-emissions as they are also re-emitted from the land surface. In this case, the combined contributions from terrestrial surface re-emissions comprised 28.2% of the annual TGM.

The ratios of natural surface to anthropogenic contributions to TGM were 67.8%/32.2% in the cold season and 77%/23% in the warm season (Fig. S5). GEM re-emission contributions comprised 26% of TGM in the warm season compared to only 2.8% in the cold season. Crustal/soil TGM contribution in the warm season was twice that of the cold season. In contrast, relative contributions from background Hg and road salt were greater in the cold season than warm season.





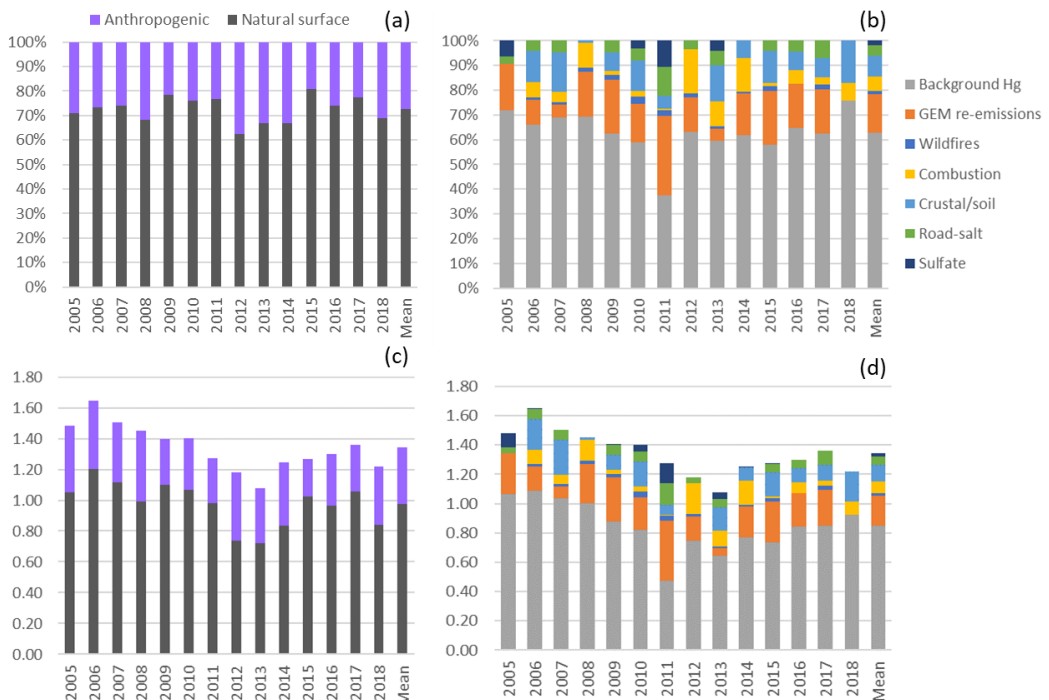

**Figure 9: Impact of natural surface emissions, anthropogenic emissions, and individual emission sources on annual TGM at Egbert (EGB). (a) and (b) plots are relative contributions; (c) and (d) are contributions expressed in concentrations (ng m⁻³). Mean applies to the 2005-2018 period. Natural surface emissions comprise wildfires, GEM re-emissions, crustal/soil dust, road salt, and natural surface emissions contributing to background Hg. Anthropogenic emissions comprise local combustion, sulfate, and anthropogenic emissions contributing to background Hg.**

### 3.2.4 Interannual and daily variability

During the 2005-2018 period, the annual relative contributions to TGM varied between 19% and 38% for anthropogenic emissions (Fig. 9). Anthropogenic contributions to TGM were especially greater during 2012-2014 because of the higher relative contributions from local combustion. The average relative contribution from local combustion was 6% during 2005-2018; the percentages during 2012-2014 were 10-18%. The annual percentage contribution to TGM during 2005-2018 reached as high as 32% for GEM re-emissions, 18% for local combustion, 17% for crustal/soil dust, 12% for road salt, 11% for secondary sulfate, and 3% for wildfire emissions. The background Hg contribution was unusually low in 2011 (37.4%) because data were mostly available during the warm season when the background impact tends to be lower. The interannual variability was typically 58-76% in other years.

Natural surface emission contributions in the warm and cold season varied in the range of 64-85% and 59-74%, respectively (Fig. S5). The warm to cold season ratio varied significantly for TGM contributions from wildfires (10 to 30) and surface emissions (5 to 35). This is because of the year-to-year variability in wildfire emissions. GEM re-emissions depend on meteorology, soil temperature and moisture, land disturbance, canopy shading, etc. which can lead to interannual variability



in the emissions. In contrast, the warm to cold season ratios were less variable for background Hg contributions (0.5 to 1) and local combustion (0.2 to 1).

Overall, emissions from natural surfaces contributed more to TGM on a daily basis compared to anthropogenic emissions;
however, there were some exceptions. Anthropogenic contributions dominated over natural surface contributions in the cold season of 2008, 2012 and 2014 (Fig. 4). While this is expected for the cold season, there were also enhanced anthropogenic daily TGM contributions during the warm season of 2012. There was a pronounced seasonal variation in background Hg contributions; the percentages were as low as 0.2% in the warm season to as high as 100% in the cold season (Fig. 10). On a daily scale, TGM contributions from GEM re-emissions or wildfire can account for more than 50% in the warm season. Daily
TGM contributions from local combustion typically comprised less than 30%, though it can occasionally exceed 50%. Secondary sulfate daily TGM contributions were mostly negligible except on a few occasions. TGM contributions from crustal/soil and road salt were significant on selected days (up to 90%).

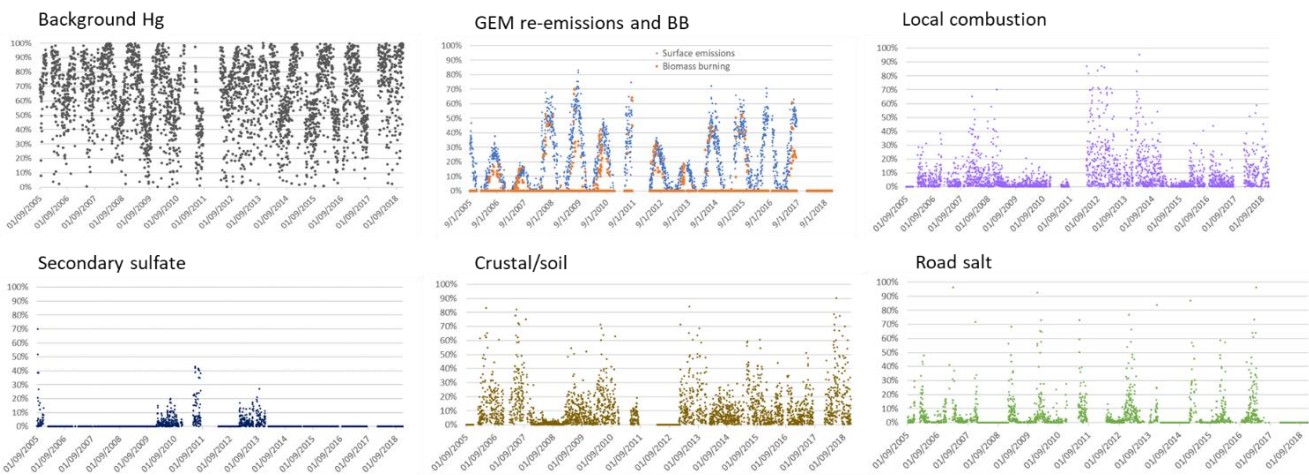

**Figure 10: Relative contributions to 24-h mean TGM from various sources at EGB**

**3.2.5 Long-term trends**

Annual observed TGM at EGB decreased at a rate of -0.026 ng m$^{-3}$ yr$^{-1}$ (p<0.001) during 2005-2018. The trend in PMF modeled TGM was similar to the observed trend (-0.027 ng m$^{-3}$ yr$^{-1}$, p<0.001; Fig. 6), which indicates good agreement between the PMF model and observations. The observed TGM trend for 2005-2018 was slightly greater than that of 1996-2010 from
the previous update (-0.02 ng m$^{-3}$ yr$^{-1}$; Cole et al., 2014). On a long-term scale, background Hg was a key driver of the observed TGM trend given the large slope of -0.017 ng m$^{-3}$ yr$^{-1}$ (Table 2). Trends in TGM contributions from other sources were small and not statistically significant. For example, TGM contributions from local combustion showed a flat trend. The major combustion sources near EGB (≤150 km) are iron and steel manufacturing and cement production (Fig. S6). Iron and steel manufacturing Hg emissions had decreased before 2009 and then increased thereafter. Hg emissions from cement production





were stable during 2006-2014 and then increased to higher levels after 2014. The change in direction of the emission trends may explain a lack of trend in TGM contributions from local combustion. The trends in relative contributions to TGM from natural surface and anthropogenic emissions were both non-significant (Fig. 11). This is expected considering the relative source contribution trends were not statistically significant or in some cases the slopes were very small.

**Figure 11: Long-term trends in relative source contributions at EGB. (a) natural surface emissions (wildfires plus re-emitted Hg), (b) anthropogenic emissions, (c) background Hg, (d) GEM re-emissions, (e) crustal/soil, (f) local combustion, (g) road salt, (h) wildfires, (i) secondary sulfate. Blue line: relative contributions; red line: trendline; green text: slope of the trendline (x100% yr$^{-1}$).**





Short term patterns in TGM were also observed, e.g. the decrease in TGM concentrations from 2005 to 2013 followed by an increase and then a flat trend. These patterns are likely influenced by local iron and steel manufacturing and cement production (Fig. S6). Hg emissions from electric utilities in the Province of Ontario decreased by 85% between 2005 and 2013 (Fig. S7c) resulting from the phase-out of coal-fired power plants. Reductions in these emissions were also seen in the U.S. Midwest and Northeast regions (Fig. S7g). The pattern in TGM after 2013 was not likely not related to local combustion. This is because while both TGM and $SO_2$ concentrations at EGB decreased during 2005-2013, there was a decoupling of TGM and $SO_2$ after 2013. The rise in TGM was likely from increased contributions from background Hg, GEM re-emissions, and crustal/soil Hg (Fig. 9).

### 3.2.6 Cold and warm season trends

Decreasing trends in the observed TGM concentrations were observed for the cold and warm seasons (Table S3). The rate of decrease was greater in the warm season (-0.030 ng m$^{-3}$ yr$^{-1}$, p=0.02) than cold season (-0.021 ng m$^{-3}$ yr$^{-1}$, p=0.02). The PMF model reproduced the seasonal trends in observed TGM. The warm and cold season modeled TGM trends were -0.028 ng m$^{-3}$ yr$^{-1}$ (p=0.02) and -0.024 ng m$^{-3}$ yr$^{-1}$ (p=0.06), respectively (Table S3). Reductions in the TGM contributions from background Hg was the main driver of the TGM seasonal trends; the slopes were -0.03 ng m$^{-3}$ yr$^{-1}$ for the warm season and -0.02 ng m$^{-3}$ yr$^{-1}$ for the cold season. The trends for TGM contributions from other sources were not statistically significant. There were also no significant changes in the relative contributions of TGM from natural surface or anthropogenic emissions for the warm and cold season (Table S4).

### 3.3 Kejimkujik National Park

### 3.3.1 Overview of TGM concentrations

The range in annual mean TGM concentrations was 1.1-1.71 ng m$^{-3}$ at Kejimkujik National Park (KEJ) during 2005-2016 (Fig. 1; Table S2). The RSD of 24 h mean TGM was 12.6% and increased year over year. The 75$^{th}$ percentile concentrations were 1.28-1.82 ng m$^{-3}$ depending on the year. Comparing the statistics over the same period (2009-2016), the mean TGM concentrations among SAT, EGB and KEJ were comparable with no statistically significant differences. The annual TGM across the three CAPMoN sites differed between 3% (in 2014) and 19% (in 2013). The 75$^{th}$ percentile was highest at SAT during 2009-2010, KEJ during 2011-2013, and EGB during 2014-2016, which shows there was no particular site experiencing higher TGM over this period. The seasonal pattern in TGM at KEJ was similar to those of SAT and EGB. Mean TGM was highest during winter and spring and lowest during summer and fall. The diel amplitudes in summer (22%) and fall (14%) were greater than those in winter (5%) and spring (8%). Hourly TGM peaked during 14:00-15:00 LST in the summer and 12:00-14:00 LST in the fall. In the winter and spring, the maximum TGM was typically observed during 11:00-13:00 LST. The TGM minimum occurred during 5:00-6:00 LST in all seasons. For the 2005-2016 period, the highest TGM was observed





in 2005 and the lowest in 2006 and 2016. Annual TGM decreased significantly from 2005 to 2006. A parabolic pattern was observed thereafter with concentrations increasing during 2006-2011 and then decreasing during 2012-2016 (Fig. 1).

### 3.3.2 PMF factor profiles

For the KEJ site, the PMF model resolved six factors. The factors represented background Hg, terrestrial GEM re-emissions/biomass burning, local combustion, secondary sulfate, aged SSA, and fresh SSA (Fig. 12). TGM (71%) and CO
(67%) were the most abundant in the background Hg factor. GEM re-emissions/biomass burning factor showed a strong presence in temperature (100%) and a weak presence in TGM (10%). Local combustion was characterized by the $SO_2$ abundance (87%); TGM (2%) and CO (2%) were not present in significant amounts. Secondary sulfate showed high abundances for sulfate (64%) and ammonium (77%) with a minor abundance for TGM (1%). Aged SSA was characterized by moderate abundances of $Ca^{2+}$, $Mg^{2+}$, $Na^+$ and $SO_4^{2-}$(22-26% for these ions) and absence of $Cl^-$. Fresh SSA comprised mostly
of $Cl^-$ (91%), $Na^+$ (65%) and $Mg^{2+}$ (60%). The TGM abundances were 10% in aged SSA and 7% in fresh SSA, which were comparable to the SAT coastal site.

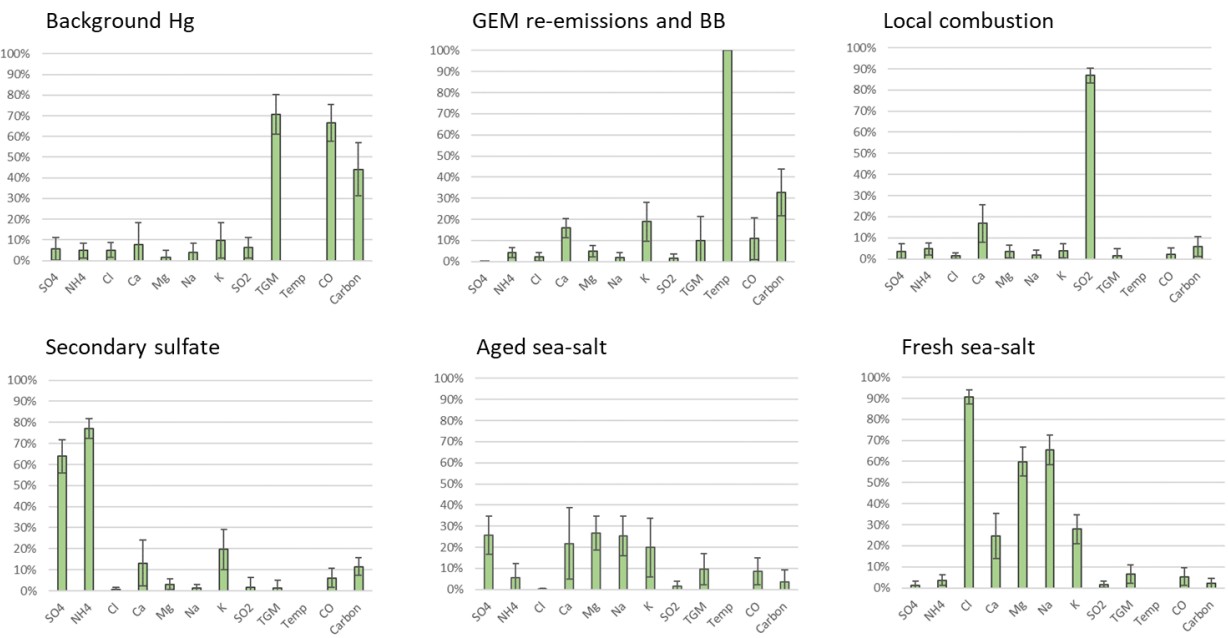

**Figure 12: PMF mean factor profiles for KEJ. Error bars indicate standard deviation among the years; BB: biomass burning.**

### 3.3.3 Overview of TGM source contributions


The mean relative contribution to annual TGM was 65% from natural surface emissions during the 2005-2016 period with the remainder from anthropogenic emissions (Fig. 13). This corresponds to average TGM contributions of 0.83 ng m$^{-3}$ and 0.45



ng m$^{-3}$, respectively. Annual TGM was strongly attributed to background Hg (71%), followed by regional emission and SSA processing (9.7%), terrestrial GEM re-emissions (8%), oceanic evasion (6.6%), wildfires (2.1%), local combustion (1.6%),

and secondary sulfate (1.4%), respectively (Fig. 13). The regional emission and SSA processing factor is considered anthropogenic origin because aged sea-salt is formed from the reaction of sea-salt and acidic gases (H$_2$SO$_4$ and HNO$_3$), whose precursors are anthropogenic SO$_2$ and NO$_x$. One of the major sources of SO$_2$ and NO$_x$ is fuel combustion from electric utilities, which are also significant sources of atmospheric Hg. There are no electric utilities within 200 km of KEJ since 2007; however, the site is frequently impacted by air masses from the northeastern U.S. region where there is a high density of electric utilities.

Thus, this factor represents both the regional emission contributions of acidic gases and Hg and subsequent chemical processing of fresh SSA as regional air masses are transported across the MBL. On the other hand, fresh SSA is directly emitted from the ocean and has not been subject to chemical processing. Hg contributions from marine air are mainly from evasion of GEM from the ocean or partitioning of GEM or GOM from sea-salt aerosols.

Emissions from natural surfaces contributed more to TGM in the warm season than the cold season, whereas the case was

reversed for anthropogenic contributions to TGM. The proportion of natural surface contributions was 63% in the cold season and 69% in the warm season (Fig. S8). Relative TGM contributions from background Hg were greater in the cold season than the warm season. In contrast, GEM re-emissions and wildfire contributions to TGM were greater in the warm season comprising 16% and 5%, respectively, of the warm season TGM.

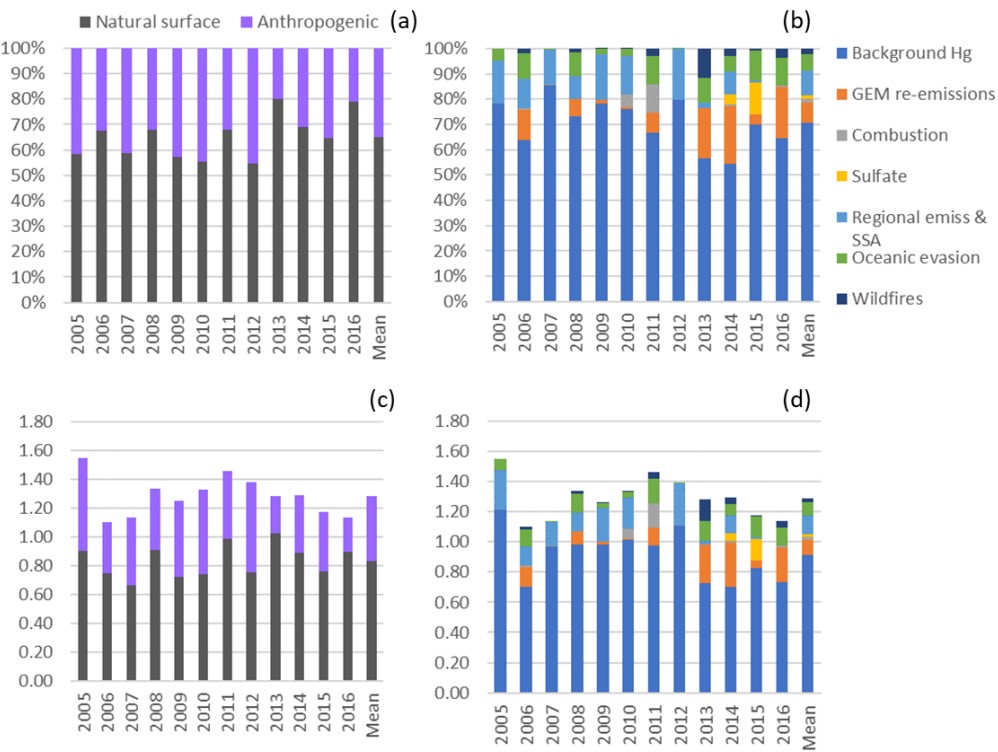





**Figure 13: Impact of natural surface emissions, anthropogenic emissions, and individual emission sources on annual TGM at Kejimkujik (KEJ). (a) and (b) plots are relative contributions; (c) and (d) are contributions expressed in concentrations (ng m$^{-3}$). Mean applies to the 2005-2016 period. Natural surface emissions comprise wildfires, GEM re-emissions, oceanic evasion, and natural surface emissions contributing to background Hg. Anthropogenic emissions comprise local combustion, sulfate, regional emissions and sea-salt processing, and anthropogenic emissions contributing to background Hg.**

### 3.3.4 Interannual and daily variability

For the 2005-2016 period, relative contributions to annual TGM ranged between 55% and 80% for natural surface emissions (Fig. 13). This percentage was highest in 2013 due to enhanced terrestrial GEM re-emissions and wildfire contributions. The anthropogenic emissions percentage was highest in 2012 because of the greater relative contribution from regional emission and SSA processing. Annual relative contributions from background Hg ranged between 55% and 85%. The annual percentage contribution to TGM during the 2005-2016 period reached as high as 23% for GEM re-emissions, 11% for local combustion, 12.5% for secondary sulfate, 20% for regional emission and SSA processing, 12% for oceanic evasion, and 11.5% for wildfires. Natural surface contributions typically exceeded anthropogenic contributions to daily TGM; however, there were also many days when the latter was equivalent or greater (Fig. 4). Anthropogenic TGM comprised 5-50% occasionally increasing above 70%. The elevated episodes were attributed to regional emission and SSA processing, local combustion, and secondary sulfate.

Daily TGM contributions from background showed large fluctuations with percentages ranging from 0% to 100%, though they were mostly above 50% (Fig. 14). GEM re-emissions and wildfires contributed upwards of 20% of the daily TGM in the warm season, whereas the contributions were negligible in the cold season. Daily TGM contributions from oceanic evasion were typically below 30% and occasionally reached as high as 90%.

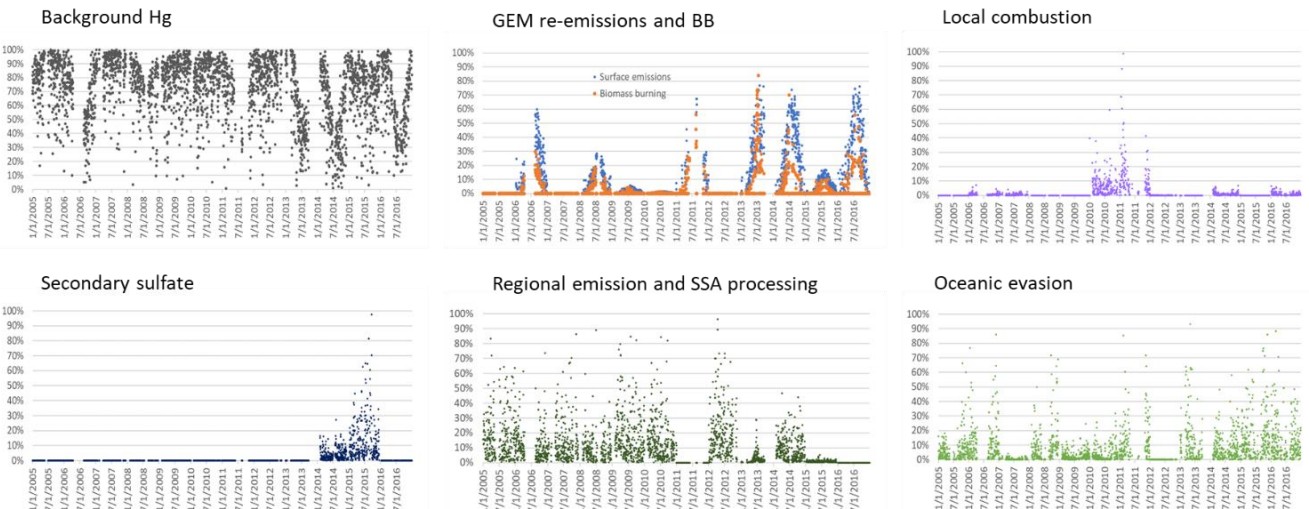

**Figure 14: Relative contributions to 24-h mean TGM from various sources at KEJ**



### 3.3.5 Long-term trends

Figure 6 shows the annual trends in the observed and modeled TGM at KEJ over the 2005-2016 period. The magnitudes of the trend were -0.014 ng m$^{-3}$ yr$^{-1}$ (1.06% yr$^{-1}$) for observed TGM and -0.015 ng m$^{-3}$ yr$^{-1}$ (1.12% yr$^{-1}$) for PMF modeled TGM.

Neither trend was statistically significant. The similarity in the trends indicate the PMF model reproduced the variability in observed TGM. The observed TGM trend for 2005-2016 were the same as that of 1996-2010 from the previous update (Cole et al., 2014). Long-term trends in TGM were driven by several sources including background Hg, regional emission and SSA processing, and GEM re-emissions. Background Hg contributions decreased significantly at a rate of -0.022 ng m$^{-3}$ yr$^{-1}$ (Table 2), which outpaced the observed TGM trend. This was also the case for the trend in regional emission and SSA processing

contributions (-0.019 ng m$^{-3}$ yr$^{-1}$). These decreasing trends were attenuated by that of GEM re-emissions contributions, which had a slight positive trend of 0.004 ng m$^{-3}$ yr$^{-1}$. The decreasing trend in the TGM contribution from regional emission and SSA processing were attributed to decreasing Hg and $SO_2$ emissions. On a regional scale, Hg emissions fell from 23.2 Mg yr$^{-1}$ in 2005 to 5.3 Mg yr$^{-1}$ in 2016 corresponding to a 77% reduction (Fig. S7g). In addition, there was also a decrease in ambient $SO_2$ at KEJ (-0.037 µg m$^{-3}$ yr$^{-1}$, p<0.001), which was driven by significant $SO_2$ emissions reductions both locally and regionally

(Fig. S9). Consequently, SSA processing also decreased with time.

The trend in relative contribution from natural surface emissions show an increase of 1.03 % yr$^{-1}$ (p<0.001) for the 2005-2016 period (Fig. 15). This trend is attributable to the steep decline in the percentage contributions from regional emissions and SSA processing (-1.13 % yr$^{-1}$), which is anthropogenic. There were also small percentage increases in natural surface emissions contributions including from terrestrial GEM re-emissions (0.3 % yr$^{-1}$, p<0.05) and oceanic evasion (0.33 % yr$^{-1}$, non-sig.).

The percentage contribution from background Hg showed a decreasing trend of -1% yr$^{-1}$ (p<0.1) similar to that in the concentrations.

While long-term TGM concentrations at KEJ are trending downwards, there are shorter term variations that are not captured in the trends analysis. TGM decreased sharply from 2005 to 2006 (Fig. 1), which was consistent with the significant decrease in local Hg emissions from electric utilities (Fig. S10). There was a period of increasing TGM from 2006 to 2011 followed by

decreasing TGM from 2011 to 2016. This pattern was consistent with local Hg emissions from oil and gas pipelines and storage emissions. These emissions began to decline in 2011. In fact, no Hg emissions within 150 km of KEJ were reported from 2014 onwards.





**Figure 15: Long-term trends in relative source contributions at KEJ. (a) natural surface emissions (wildfires plus re-emitted Hg),**
**(b) anthropogenic emissions, (c) background Hg, (d) regional emission and SSA processing, (e) oceanic evasion, (f) GEM re-**
**emissions, (g) wildfires, (h) local combustion, (i) secondary sulfate. Blue line: relative contributions; red line: trendline; green text:**
**slope of the trendline (x100% yr⁻¹).**

### 3.3.6 Cold and warm season trends

The PMF model reproduced the observed TGM trends for the cold and warm seasons. The observed TGM decreased by -
0.0232 ng m$^{-3}$ yr$^{-1}$ and -0.0199 ng m$^{-3}$ yr$^{-1}$ in the cold and warm season, respectively (Table S3). The corresponding trends in
the modeled TGM were -0.0224 ng m$^{-3}$ yr$^{-1}$ and -0.0172 ng m$^{-3}$ yr$^{-1}$, respectively. Negative TGM trends in the cold season were
driven by decreasing trends in regional emissions and SSA processing with a slope of -0.0183 ng m$^{-3}$ yr$^{-1}$ (p=0.08). Background
Hg contributions declined at a faster rate in the warm season (-0.0553 ng m$^{-3}$ yr$^{-1}$, p<0.05) compared to the cold season. Yet
the observed TGM trend in the warm season was slower than that of the cold season. The reason is likely because of the



positive trends in GEM re-emissions, wildfires, and oceanic evasion in the warm season that partially offset the large negative trends in background contributions.

Relative TGM contributions from natural surfaces were not statistically significant in either season (Table S4). The overall decreasing trend in the relative contribution from background Hg was driven by the warm season trend with a slope of -3 % yr$^{-1}$ (p=0.01). The overall increasing trends in GEM re-emissions and wildfires percentage contributions were due to their

strong warm season trends.

## 4 Discussion

TGM source apportionment results were compared among the sites. Natural surface emissions dominated anthropogenic emission contributions to TGM at the three sites. Mean relative contributions from anthropogenic emissions were both 35% at SAT and KEJ and lower at EGB. There was a clear long-term increase in the relative importance of natural surface emissions

to TGM at SAT and KEJ, which resulted from decreased anthropogenic Hg emissions and increased oceanic Hg evasion and terrestrial GEM re-emissions.

Background Hg derived from the northern hemispheric Hg pool was a major source of TGM at the three sites. Average relative contributions to annual TGM were 71% at KEJ, 63% at EGB, and 53% at SAT, with higher contributions from November to April. The results at CAPMoN sites are similar to the baseline factor identified at a remote monitoring site in Mace Head,

Ireland. In that study, the baseline factor was also the dominant Hg source impacting TGM (Custodio et al., 2020). Source attribution analysis performed using the Global/Regional Atmospheric Heavy Metals (GRAHM) model showed that the sources of GEM in Canada mostly originated in Europe and East Asia. European sources contributed 3% of GEM at SAT, <3% at EGB, and 3.5% at KEJ via long-range transport across the Arctic. Long-range transport from East Asia had the greatest impact on GEM across Canada, contributing 16% at SAT, 14-15% at EGB, and 15% at KEJ (ECCC, 2016). The anthropogenic

to natural surface emissions proportion for background Hg was estimated based on global Hg emissions inventory. While the percentages used in this study were consistent with some emissions inventories, i.e. ~30% for anthropogenic Hg emissions (Pirrone et al., 2010; Outridge et al., 2018; Streets et al., 2019; UNEP, 2019), this percentage is 4-6% lower in other global budgets (Shah et al., 2021; Sonke et al., 2023; Zhang et al., 2023). It is expected that the global anthropogenic to natural contributions ratio will change in the future as primary anthropogenic emissions decrease and Hg re-emissions increase.

Terrestrial surface re-emissions was the second most important Hg source at EGB (28%) and SAT (14%). Relative contributions from GEM re-emissions were greatest from May to October. GEM re-emission was only a minor source of TGM at KEJ likely because the site is in a forested area which is a net sink for GEM. The variability between sites reflects the nature of GEM surface-air exchange, which is spatially variable and dependent on factors, such as solar radiation intensity, canopy shading and wetness, air and soil temperature, soil Hg content, soil moisture, soil organic matter content, vegetation and

litterfall covering the soil, ambient Hg concentration, and forest uptake which is a key driver of GEM dry deposition (Zhang et al., 2009; Ottesen et al., 2013; Agnan et al., 2016; Eckley et al., 2016; Wang et al., 2016; Zhu et al., 2016; Sommar et al.,



2020). Some areas of British Columbia and Ontario have soils that are naturally enriched in Hg, which have led to higher fluxes (ECCC, 2016). Hg emissions from agricultural soils (e.g. at EGB) are greater than that of natural soils because of more frequent land disturbance which increases volatilization and soil dust resuspension (Cobbett and Van Heyst, 2007; Zhu et al.,

2016). Surface re-emission is one of the least constrained processes in the Hg cycle. It can be influenced by a multitude of environmental factors, which are currently not well represented in CTMs (Zhu et al., 2016; Obrist et al., 2018). There are also limited Hg flux measurements for characterizing both spatial and temporal variations.

TGM contribution from local combustion was highest at EGB (6%) followed by SAT (4%) and KEJ (2%). These results were similar to previous GRAHM model simulations, which found Canadian sources accounted for only 3-5% of GEM near

EGB, 1% near SAT, and 0.5-1% near KEJ (ECCC, 2016). Overall, the contribution by Canadian sources is typically < 1% in areas outside the vicinity of major Hg point sources according to models (ECCC, 2016).

For secondary sulfate TGM contributions, the percentages ranked by site were SAT > EGB > KEJ and were under 5%.

Oceanic evasion contributed similar percentages of the TGM at SAT and KEJ (7-9.5%). Recent model simulations suggest that Hg re-emission from oceans is underestimated by 40% (Zhang et al., 2023). Oceanic evasion impacts to coastal sites and

the global Hg pool may be greater than previously thought.

At the coastal sites, we found evidence of anthropogenic emission contributions and SSA processing. Marine transportation and shipping ports provided sources of TGM at SAT, while regional Hg emissions notably from U.S. electric utilities (fossil fuel combustion) had impacted TGM at KEJ. Because these sources also emit $SO_2$, the acidic gases formed further reacted with SSA in the MBL resulting in the formation of aged SSA. Significant reductions in both Hg and $SO_2$ emissions led to

diminished impacts from anthropogenic emissions and SSA processing over time.

Wildfires affected TGM at SAT (western Canada) more than KEJ and EGB (eastern Canada). Wildfires in western Canada and the western U.S. have led to elevated $PM_{2.5}$, $O_3$, atmospheric nitrogen and total carbon in downwind regions (Lu et al., 2016; McClure and Jaffe, 2018; Chen et al., 2019; Campbell et al., 2022). Wildfires are also an important source of Hg in western Canada, comprising approximately 65% of the national total wildfire Hg emissions. For the 2010-2015 period,

wildfires contributed 0.2-0.4% of the daily mean GEM near SAT according to the GEM-MACH-Hg model (Fraser et al., 2018). In this study, we estimate the PMF-derived daily mean TGM contribution from wildfires to be 4.4% over the same period. For EGB, the percentage wildfire contributions to daily mean GEM/TGM were 0.4-0.8% based on the GEM-MACH-Hg model and 1.7% based on the PMF model. The corresponding percentages for KEJ were 0.3-0.5% and 2.7% of the daily mean GEM/TGM. Note that on a daily basis there can be large variability in the GEM-MACH-Hg estimated wildfire

contributions with percentages up to 30% for SAT, 23% for EGB and 10% for KEJ. This variability was also seen in the PMF modeling results (Figs. 5, 10, 14). Differences in the wildfire source contributions between GEM-MACH-Hg (Fraser et al., 2018) and PMF (this study) could be due to underestimated wildfire Hg emissions in CTMs, other model




parameterization uncertainties, measurement uncertainties affecting the PMF model results, FRP-approach for screening wildfire TGM contributions, etc. While model intercomparisons are common for CTM, there needs to be more comparisons

conducted between CTM and receptor models like PMF which will ultimately improve source apportionment estimates for both types of air quality models. We examined wildfire impacts on gaseous Hg in this study; however, wildfires also contribute significantly to particulate Hg depending on the fuel moisture and combustion type (Obrist et al., 2008; McLagan et al., 2021). To capture the full impact of wildfires on atmospheric Hg, the PMF analysis will need to include particulate Hg measurements.

**5 Conclusions**

Source contributions to three rural-remote TGM sites were estimated using the PMF model. We examined long-term trends in TGM source contributions to understand the major drivers of the observed TGM trends, as well as analyzed the variability in interannual, seasonal and daily TGM contributions. Anthropogenic sources that were inferred from PMF include local combustion, secondary sulfate, and the anthropogenic portion of background Hg. Additionally, shipping emissions were

identified at SAT, and regional emissions from the U.S. northeast were identified at KEJ. Natural surface emission contributions comprised terrestrial GEM re-emissions, oceanic Hg evasion, wildfires, and the proportion of background Hg from natural surface emissions. Additional TGM contributions from crustal/soil dust and road salt were found at EGB. At SAT, the decreasing TGM trend was attributed to decreasing shipping and GEM re-emissions. The long-term decreasing TGM trend at EGB was attributed to decreasing background Hg contributions, though other local sources contributed to the initial

decline in TGM and the flat trend after 2013. The long-term TGM trend at KEJ was driven by background Hg, regional emissions, and GEM re-emissions. When analyzing on a shorter timescale, Hg emissions from electric utilities and oil and gas pipelines and storage had a strong influence on TGM. Overall, emission contributions from natural surfaces dominated anthropogenic contributions to annual TGM at the three sites.

In the last decade, the downward trend in anthropogenic emission contributions led to increasing Hg contributions from natural

surfaces. The latter can potentially be accelerated by global warming as this can drive up terrestrial Hg re-emissions, oceanic Hg emissions, and wildfire Hg emissions. These emissions not only contribute to downwind areas, but also to the global Hg pool, which in the case of GEM impacts other regions through long-range transport. Hg emissions from natural surfaces are overall less constrained compared with combustion sources, and they make up the bulk of the global Hg emissions. It is important to increase monitoring of surface re-emissions, track their long-term trends, and examine how climate perturbations

are affecting the emissions and Hg cycling in the environment.

**Data availability:** All datasets used in this study are publicly available (Table S1).

**Competing interests:** One of the authors is a member of the editorial board of Atmospheric Chemistry and Physics.



**Acknowledgements:** The first author acknowledges ECCC Canadian Air and Precipitation Monitoring Network (CAPMoN),
National Air Pollution Surveillance (NAPS) program, Canadian Greenhouse Gas Measurement program, Climate Data
Services; USEPA Air Quality System; Interagency Monitoring of Protected Visual Environments (IMPROVE); and
NASA/MODIS for the provision of datasets used in this publication. Special thanks to CAPMoN and National Atmospheric
Chemistry (NAtChem) teams; Amy Hou for data extraction/processing; Doug Worthy for Egbert CO data; Anne Marie
Macdonald, James Kuchta and Kenny Yan for internal review.

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
