# Peer review of "Natural Surface Emissions Dominate Anthropogenic Emissions Contributions to Total Gaseous Mercury at Canadian Rural Sites"

_EGUsphere, 2024_

## Author Comment (AC1)

**Authors' Response to Reviewers**

**By Irene Cheng, Amanda Cole, Leiming Zhang, and Alexandra Steffen (Environment and Climate Change Canada)**

**Reviewer 2**

We appreciate the comments from this reviewer and have provided responses to all comments and improved on the manuscript.

Cheng et al. performed a thorough analysis of TGM from three Canadian sites using PMF. I have a couple of concerns. First, I am not entirely convinced how some of the factors were identified, i.e., the background factor and the reemission/biomass burning factor. The background factor was identified because of the high abundance of CO and TGM, and the authors cited Weiss-Penzias et al. (2007) to back the decision. What was their definition of "background"? I took it as the baseline level at the site. Did they look into the correlation between CO and TGM? Chances are the two are correlated due to their similar seasonal patterns. CO has always been used as an anthropogenic tracer in the literature. The very reference, Weiss-Penzias et al. (2007), they cited used the CO-TGM correlation to demonstrate the impact of Asian pollution. Therefore, the authors' decision to use CO as a background tracer did not make sense to me. Numerous studies in the literature used TGM-CO correlation at sites to identify anthropogenic influence; yet the two compounds do not really share common sources. There is in fact a deeper meaning to this correlation, which is, in this reviewer's opinion, that the relationship really reflects the anthropogenic or wildfire burning emission ratios of TGM and CO over a studied region. Regarding the reemission/biomass burning factor, there were quite jarring inconsistencies in the concentrations of supposedly fire tracers. As is commonly known, biomass burning emissions can enhance CO, TGM, and $K^+$. However, Fig. 2 showed $K^+$ at ~10%, similar to the values for the local combustion and fresh SSA factors and lower than the $K^+$ values (~15%) in the background, secondary sulfate and aged SSA factors, and also showed the lowest CO in this factor.

Response: In the revised paper, the 'background' factor was renamed to the Hg pool, which was discussed in section 2.2 and 3.1.2. This factor encompasses natural, anthropogenic, and re-emitted Hg mostly from the northern hemisphere that is subject to long range transport. It is not associated with any specific point sources, and CO is not being used as an anthropogenic tracer. As this reviewer noted, the Hg-CO relationship may be indicative of anthropogenic or biomass burning. CO is an indicator of the Hg pool and long-range transport because its lifetime is in the order of a few months, which allows for it to be transported over long distances (Jeffery et al., 2024). Given the relatively small magnitude of combustion sources near these rural and remote sites, it is not surprising that the CO emissions from these sources contribute a relatively small proportion of the total CO observed, which is dominated by the hemispheric background. This characteristic is similar to that of GEM and TGM. This has been clarified in the revised paper (section 3.1.2), which reads "The Hg pool was identified based on the high abundance of CO (56%). The Hg pool consists of natural, anthropogenic, and re-emitted Hg mostly from the Northern Hemisphere that is subject to long range transport (Selin and Jacob, 2008). CO is emitted from fossil fuel combustion and wildfires. It has a longer lifetime in the order of a few months compared with $SO_2$, which allows for it

to be transported long distances by advection (Jeffery et al., 2024) and accumulate in the hemispheric background. This characteristic is similar to that of GEM and TGM."

Based on monthly mean TGM contributions from the Hg pool, the seasonal patterns are more in line with anthropogenic emissions associated with winter heating across the northern hemisphere. TGM contributions from the Hg pool show a conspicuous seasonal cycle with a peak in the colder months and minimum in the warmer months (see graphs below). At SAT, the range in TGM contributions for the cold and warm season are 0.75-0.93 ng m$^{-3}$ and 0.42-0.74 ng m$^{-3}$, respectively. The range refers to the interannual variability. At EGB, the ranges were 0.67-1.35 ng m$^{-3}$ for the cold season and 0.47-0.93 ng m$^{-3}$ for the warm season. At KEJ, the ranges were 0.86-1.28 ng m$^{-3}$ for the cold season and 0.38-1.03 ng m$^{-3}$ for the warm season. If this factor was driven by wildfires, the warm (fire) season contributions would have been greater than those of cold season. Previous chemical transport modeling studies using the GEM-MACH-Hg model show only a small Hg contribution from wildfires at these sites (Fraser et al., 2018).

[Figure]

[Figure]

[Figure]

Hg re-emissions and wildfires were identified from a factor with a strong temperature signal (Figures 2, 8, and 12). This reviewer noted that Fig. 2 showed the $K^+$ percentage at ~10% in this factor, which is similar to the percentages in other factors. This may be because this factor is only partially due to wildfires, and re-emissions of GEM would not be associated with $K^+$. As well, $K^+$ can also be released from crustal dust, soil and vegetation (Zhang et al., 2008) and is very common in pollen (Lee et al., 1996). Considering the site in question is coastal, the source of $K^+$ may include sea-salt aerosols. Potassium is also associated with coal combustion (Yu et al., 2018), although this is less likely the case for the rural-remote sites in this study. Thus, the sources of $K^+$ are wide-ranging, and it is not necessarily a strong indicator of wildfires especially if the wildfire area is far away from the monitoring site. This discussion was added to the revised paper (section 3.1.2), which reads "The $K^+$ percentage for biomass burning and re-emissions was comparable to those in other factors. This may be because this factor is only partially due to wildfires, and re-emissions of GEM would not be associated with $K^+$. As well, $K^+$ can also be released from the continental crust, soil and vegetation (Zhang et al., 2008), and is very common in pollen (Lee et al., 1996). Since SAT is a coastal site, the source of $K^+$ may include SSA. Potassium is also associated with coal combustion (Yu et al., 2018), although this is less likely the case for the rural-remote sites in this study." The novel approach that was applied in this study was using Fire Radiative Power (FRP) satellite data to screen wildfire contributions to TGM. This approach does not rely solely on presumed fire tracers; it also accounts for concurrent wildfire observations in the region surrounding each site. We examined correlations between observed $K^+$ concentrations and FRP on a daily and monthly scale for the SAT site. No strong relationships were found ($r^2$ of 0.0003 daily and 0.049 monthly), suggesting that $K^+$ was not the best fire tracer for the studied sites.

Fraser, A., Dastoor, A., and Ryjkov, A.: How important is biomass burning in Canada to mercury contamination?, Atmos. Chem. Phys., 18(10), 7263-7286, 2018.

Jeffery, P. S., Drummond, J. R., Zou, J., and Walker, K. A.: Identifying episodic carbon monoxide emission events in the MOPITT measurement dataset, Atmos. Chem. Phys., 24, 4253–4263, https://doi.org/10.5194/acp-24-4253-2024, 2024.

Lee, E. J., Kenkel, N., and Booth, T.: Atmospheric deposition of macronutrients by pollen in the boreal forest, Ecosci., 3(3), 304-309, 1996.

Yu, J., Yan, C., Liu, Y., Li, X., Zhou, T., and Zheng, M.: Potassium: a tracer for biomass burning in Beijing?, Aerosol Air Qual. Res., 18(9), 2447-2459, 2018.

Zhang, L., Vet, R., Wiebe, A., Mihele, C., Sukloff, B., Chan, E., Moran, M. D., and Iqbal, S.: Characterization of the size-segregated water-soluble inorganic ions at eight Canadian rural sites, Atmos. Chem. Phys., 8, 7133–7151, https://doi.org/10.5194/acp-8-7133-2008, 2008.

Second, what does the "secondary sulfate factor" really mean? If I am not mistaken, it could indicate the role of secondary production of TGM. If it was correct, then the assumption of chemistry being negligible would be invalid. I'm curious how such a paradox can be reconciled.

Response: The secondary sulfate factor is indicative of regional Hg emissions and chemical transformation including oxidation and gas-particle partitioning. This is because the strong sulfate presence in a factor has been associated with regional emissions and oxidation of $SO_2$ from previous source apportionment studies (Liu et al., 2003; Keeler et al., 2006; Gratz and Keeler, 2011; Pancras et al., 2013). In previous studies, back trajectory analyses were produced for the studied sites (except SAT because of modeling over complex terrain) to estimate the fractional contributions of wind direction sectors to atmospheric sulfate deposition. The analyses showed the EGB and KEJ sites were frequently impacted by regional transport. There were regional contributions from the south-southwest sectors and west-northwest sectors for the EGB site and from the southwest and northwest sectors for the KEJ site (ECCC, 2004; Zhang et al., 2008). Additional explanation on the secondary sulfate factor were included in the revised paper (section 3.2.2), which reads "The secondary sulfate factor is indicative of regional Hg emissions and chemical transformation including oxidation and gas-particle partitioning. This is because the strong sulfate presence has been associated with regional emissions and oxidation of $SO_2$ from previous source apportionment studies (Liu et al., 2003; Keeler et al., 2006; Gratz and Keeler, 2011; Pancras et al., 2013). Previous back trajectory analyses showed the EGB site was frequently impacted by regional transport (ECCC, 2004; Zhang et al., 2008)."

The PMF model assumes a constant source profile, which may be violated in the case of chemical species undergoing transformation during transport from the source to receptor site, e.g. using gaseous elemental Hg (GEM) or gaseous oxidized Hg (GOM or RGM) data. As stated in section 2.2, TGM is the sum of GEM and GOM, which tends to be less impacted by chemical reactions. We have also limited the input of reactive or secondary species into PMF where possible; ozone and nitrate are also monitored at the sites but were not included in the PMF model. $SO_2$ and sulfate are necessary tracers to distinguish between local combustion vs. regional emissions and fresh vs. processed sea salt.

Environment and Climate Change Canada (ECCC): 2004 Canadian Acid Deposition Science Assessment, Meteorological Services of Canada, http://www.publications.gc.ca/pub?id=9.688243&sl=0, 2004.

Gratz, L. E., and Keeler, G. J.: Sources of mercury in precipitation to Underhill, VT, Atmos. Environ., 45(31), 5440-5449, 2011.

Keeler, G. J., Landis, M. S., Norris, G. A., Christianson, E. M., and Dvonch, J. T.: Sources of mercury wet deposition in eastern Ohio, USA, Environ. Sci. Technol., 40(19), 5874-5881, 2006.

Liu, W., Hopke, P. K., Han, Y. J., Yi, S. M., Holsen, T. M., Cybart, S., Kozlowski, K., and Milligan, M.: Application of receptor modeling to atmospheric constituents at Potsdam and Stockton, NY, Atmos. Environ., 37(36), 4997-5007, 2003.

Pancras, J. P., Landis, M. S., Norris, G. A., Vedantham, R., and Dvonch, J. T.: Source apportionment of ambient fine particulate matter in Dearborn, Michigan, using hourly resolved PM chemical composition data, *Sci. Total Environ.*, *448*, 2-13, 2013.

Zhang, L., Vet, R., Wiebe, A., Mihele, C., Sukloff, B., Chan, E., Moran, M. D., and Iqbal, S.: Characterization of the size-segregated water-soluble inorganic ions at eight Canadian rural sites, Atmos. Chem. Phys., 8, 7133–7151, https://doi.org/10.5194/acp-8-7133-2008, 2008.

The manuscript is quite tedious to read. The approach is mechanical. The interpretation of the analysis results is somewhat arbitrary. There is potential to this study, but the authors might want to put more effort into thinking through the interpretation of their results.

Response: The manuscript text has been revised and condensed with greater emphasis on the interpretation of the PMF factors and major results of the study. We have provided clarification on using CO as a tracer of the Hg pool and long-range transport (section 3.1.2) and discussions on the secondary sulfate factor and additional $K^+$ sources to explain the presence of the ion in other PMF factors (section 3.2.2).

Few source apportionment studies have estimated the relative contributions of sources to TGM over different time scales including annual, seasonal, daily, and on a long-term basis. These results, while detailed and statistical in nature, may be of interest to other readers. We have refined the long-term trends section to focus on the main causes of the long-term annual TGM concentrations, which is a fundamental policy question of the Minamata Convention on Mercury. Additional discussions on long-term TGM source contribution trends for the cold and warm seasons were moved to Supplement S4.

---

## Author Comment (AC2)

**Authors' Response to Reviewers**

**By Irene Cheng, Amanda Cole, Leiming Zhang, and Alexandra Steffen (Environment and Climate Change Canada)**

**Reviewer 1 (Danilo Custódio)**

We appreciate the comments from this reviewer and have provided responses to all comments and improved on the manuscript.

The use of unsupervised methods, such as Positive Matrix Factorization (PMF), in the source apportionment of Total Gaseous Mercury (TGM) has proven to be highly insightful. PMF provides a data-driven, top-down approach to estimate mercury fluxes by breaking down the observed concentrations into potential sources and their relative contributions. The strength of PMF lies in its ability to reveal hidden patterns in large datasets without predefined assumptions about the sources. This makes it a powerful tool for identifying unknown or unexpected sources of mercury emissions and for quantifying their impacts on the environment.

However, despite its advantages, the PMF model—especially as implemented in the U.S. EPA's PMF software—has some significant limitations. One of the main drawbacks is that it operates as a black box, where users have limited control over the relationships between the variables loaded into the model. While this makes the tool remarkably user-friendly and accessible to non-experts, it can also introduce risks when interpreting the results. Since users have little insight into the inner workings of the model, there is a danger of not fully understanding the factors driving the apportionment. This becomes particularly critical in environmental science, where subtle changes in the data can lead to vastly "different source contributions".

Response: We hear your concerns about the PMF model. For meeting the objectives of this study, we consider the PMF model to be the most appropriate choice for the following reasons. First, the model identifies potential sources and quantifies their contributions to TGM concentrations using in-situ measurements. Second, the TGM source contributions can be analyzed over annual, seasonal, and daily time scales to gain a deeper understanding of source impacts. In this study, we are particularly interested in understanding the emission drivers of long-term TGM trends, which is a fundamental question of the Minamata Convention on Mercury. Third, the PMF model has existed for more than 20 years with hundreds of publications using the model for source apportionment. The principles and fundamental equations are detailed in the PMF user's guide (Norris et al., 2014) and scientific literature. The PMF model ingests a dataset of observations and estimates the species' composition in each factor and the factor contributions with the objective of minimizing the scaled residuals (model-observation differences). The user must derive the optimal number of factors in the final solution by examining the model performance statistics and the physical meaning of the factors. Caution must be taken in the interpretation of the factors. Although PMF is relatively less complex than chemical transport models, there is a lot of effort involved with the measurement data preparation, running the model, analyzing and interpreting the model results, and re-tuning the model runs based on results obtained in order to come up with the optimal

solution. The interpretation of the PMF results is not trivial and should be based on scientific literature and sound knowledge of potential sources of pollution and pollutant tracers.

The main PMF drawback is not the limited control over the relationships between the variables loaded into the model. The relationships between variables can be examined prior to running the model itself. Scatterplots between the loaded variables can be viewed within the PMF interface, which allows users to examine whether there are any bivariate relationships (Norris et al., 2014).

Norris, G., Duvall, R., Brown, S., and Bai, S.: EPA Positive Matrix Factorization (PMF) 5.0 Fundamentals and User Guide. Office of Research and Development, Washington, DC 20460. https://www.epa.gov/air-research/epa-positive-matrix-factorization-50-fundamentals-and-user-guide, 2014.

Moreover, a key issue with PMF is that it will always produce an output, regardless of the quality or representativeness of the input data. This brings about the risk of "garbage in, garbage out." If the input data is not carefully curated, or if the underlying assumptions about the sources and their relationships are flawed, the model can generate misleading or incorrect apportionments. This is particularly concerning in mercury studies, where TGM concentrations are influenced by various factors, including natural emissions, re-emissions, and anthropogenic activities. The complexity of mercury's behavior in the atmosphere makes it imperative for researchers to critically assess the outputs of PMF, and not blindly trust the results.

Response: We agree that the PMF results are strongly dependent on the input data. In this study, we only used quality controlled measurement data that are published online by their respective monitoring networks. The quality control and quality assurance procedures are available from ECCC Canadian Air and Precipitation Monitoring Network (CAPMoN), ECCC National Air Pollution Surveillance (NAPS) program, ECCC Canadian Greenhouse Gas Measurement program, USEPA Air Quality System (AQS), Interagency Monitoring of Protected Visual Environments (IMPROVE), and ECCC Climate Data. In addition to using data that have been reviewed rigorously, we thoroughly checked our data processing steps for data averaging and merging. From our experience, these errors can lead to erroneous PMF outputs. After correcting errors in the input data and re-running the model, the model output was significantly more reliable. Often, the selection of chemical species is based on the measurements available at the site. We agree that TGM concentrations are influenced by different activities and sources, and this is the most challenging part of understanding atmospheric mercury processes. Therefore, in this study, we included variables that are representative of different emission sources to better derive our conclusions. This ensures the source attribution is not limited to well-established anthropogenic Hg sources (like coal combustion, metal smelting), but also the less studied Hg sources like wildfires, Hg reservoir, and terrestrial and aquatic surface re-emissions which also deserve more attention. We also recognize that chemical transformation is not taken into account in the PMF model. Therefore, we have limited the number of reactive species into the PMF model except for the ones crucial for differentiating local combustion ($SO_2$) and regional emissions transport ($SO_4^{2-}$). These chemical species are frequently used in the PMF literature. There is a lot of work that goes into preparing the dataset for PMF modeling. The steps taken in this study ensure the PMF results are robust and consider all types of Hg sources where possible.

Therefore, while PMF serves as a valuable tool for source apportionment, particularly in a top-down framework for estimating mercury fluxes, it must be used with caution. Researchers should be aware of the potential pitfalls and ensure that the input data is thoroughly vetted. Additionally, incorporating other methods to validate PMF results could mitigate the risks of misinterpretation. This reflective and cautious approach will ensure that PMF's insights into TGM source apportionment remain robust and scientifically sound.

Response: See response to the previous comment regarding the input data. Additionally, the PMF results were verified independently using source apportionment results from chemical transport modeling studies (ECCC, 2016; Fraser et al., 2018). PMF and chemical transport models (CTMs) can both produce source apportionment estimates and their methods are very different. The TGM source contributions from the PMF model were consistent with the GRAHM model (ECCC, 2016) and GEM-MACH-Hg model (Fraser et al., 2018) as discussed in section 4 of the paper. Both PMF and CTMs suggest anthropogenic TGM contributions comprised only a few percent of the annual TGM and the largest proportion of the TGM was attributed to long range transport. Wildfire contributions from the PMF model were a few percent higher compared to GEM-MACH-Hg model. However, given the uncertainties in air quality models and the extreme temporal and spatial variability of wildfires, differences in the order of a few percent are reasonable.

Environment and Climate Change Canada (ECCC): Canadian mercury science assessment: report. Gatineau, Quebec. ISBN 978-0-660-04499-6. www.publications.gc.ca/pub?id=9.810484&sl=0, 2016.

Fraser, A., Dastoor, A., and Ryjkov, A.: How important is biomass burning in Canada to mercury contamination?, Atmos. Chem. Phys., 18(10), 7263-7286, 2018.

In the source apportionment presented by the authors, several systematic issues have been identified, ranging from variable selection to the model run setup. One notable concern is the inclusion of temperature as a variable alongside atmospheric tracers. While temperature is a fundamental environmental parameter, its use in this context introduces a significant risk of spurious correlations due to its high amplitude variations throughout the year. These fluctuations can heavily influence the eigenvector decomposition in the rotational factorization, constraining the results based on seasonal temperature trends rather than genuine emission sources.

The problem arises when temperature, with its pronounced seasonal patterns, overwhelms the underlying relationships between the atmospheric tracers and the sources of mercury emissions. This can lead to misleading factor identifications, where the eigenvectors are more reflective of the temperature's seasonal cycle than of the true emission dynamics of the tracers. When eigenvectors are primarily driven by trends or seasonality, there is a high risk that the source apportionment is dictated by external variables not directly linked to the emission processes themselves. As a result, the interpretation of the PMF output can become compromised, leading to inaccurate conclusions about source contributions.

Response: The decision to include temperature in PMF is based on our understanding of GEM surface re-emissions (volatilization), which is a process strongly dependent on air temperature as well as soil properties. This was discussed in section 4 with relevant papers cited. Unfortunately, daily data on soil properties are not available at the sites. We also note that temperature was

included in the PMF model for identifying GEM surface re-emissions in a previous study (Qin et al., 2020).

Hg sources, such as wildfire emissions and terrestrial surface re-emissions, are closely linked to seasonal temperature trends. The occurrence of wildfires and GEM re-emissions are expected to increase during the warmer months. These emissions are coupled with seasonal temperature cycle. Thus, temperature should not be viewed as an external variable; it drives many of the Hg emissions. Justification for including temperature in the PMF model is provided in the revised paper (section 2.2), which reads "Temperature was included in the model because it is an important driver of wildfire emissions and terrestrial Hg re-emissions (Zhu et al., 2016)."

Qin, X., Zhang, L., Wang, G., Wang, X., Fu, Q., Xu, J., Li, H., Chen, J., Zhao, Q., Lin, Y. and Huo, J.: Assessing contributions of natural surface and anthropogenic emissions to atmospheric mercury in a fast-developing region of eastern China from 2015 to 2018, Atmos. Chem. Phys., 20(18), 10985-10996, 2020.

Furthermore, it is crucial to emphasize the importance of sensitivity tests and residual analysis to ensure the robustness of the PMF solution. Sensitivity tests help to evaluate how the model responds to changes in variable selection and model parameters, offering insights into the stability and reliability of the source apportionment. Residual analysis, on the other hand, provides a valuable check on the quality of the model fit, indicating whether the factors identified by the PMF model adequately explain the observed data or if there are unexplained variances that need further investigation.

Response: The model fit results were shown in Supplement Figure S1 and discussed in section 2.2. From Fig. S1, it can be seen that the source apportionment results are reliable. The coefficient of determination ($R^2$) between the PMF modelled and observed TGM concentrations is above 0.7 for the entire time series for each site, indicating a good model fit based on the current variable selection and model parameters chosen. There is also strong overlap between the modelled and observed time series for 24 h mean TGM, indicating the model adequately captured the daily variability. One of the top considerations in deriving the final PMF solution is the justification of the factors and assignment to Hg sources. The sensitivity tests using 5 and 7 factors resulted in factors that were difficult to interpret. A 5-factor solution results in tracers loading on multiple factors. This leads to a scenario where multiple factors can be assigned to the same source. An ideal PMF solution is one where each factor is assigned to a unique source. A 7-factor solution contains all the results of the 6-factor solution; however, it includes an additional factor that cannot be definitively assigned to a source. Results from the PMF sensitivity runs are discussed in Supplement section S2 and corresponding Tables S5-S7.

The PMF residuals are within the recommended limit of three standard deviations and follow a normal distribution (Hopke et al., 2023). This confirms the modeled factors adequately explain the observed TGM data. The strong $R^2$ between modeled and observed TGM concentrations also indicates a good model fit (Fig. S1). The model could not reproduce a few elevated TGM concentrations at the KEJ site. There was a total of 7 data points in the entire 2005-2016 time series (out of 3118 data points) where the scaled residuals were beyond three standard deviations. The residuals analysis is discussed in Supplement section S3 and corresponding Fig. S12.

Hopke, P. K., Chen, Y., Rich, D. Q., Mooibroek, D., and Sofowote, U. M.: The application of positive matrix factorization with diagnostics to BIG DATA, Chemometrics and Intelligent Laboratory Systems, 240, 104885, 2023.

I find it puzzling why the authors chose to perform a separate PMF run for each individual year, as this approach undermines the potential insights that could be gained from analyzing the full, continuous time series together. Running the entire time series as a single dataset would provide a more robust and comprehensive analysis, allowing the model to capture long-term trends, interannual variability, and potential seasonality in a more holistic way. A year-by-year analysis may artificially constrain the factors identified, leading to fragmented or incomplete source apportionment, and it limits the ability to understand how certain sources or processes evolve over time.

Response: The reviewer's point is valid; there is some loss of statistical power by not using the entire time series in a PMF run. On the other hand, there is additional information gleaned from the separate annual runs. For example, the fact that the same factor profiles are reproduced each year provides additional confidence in the results, while the variability from year to year gives us an indication of the uncertainty in those profiles. As shown by the error bars in the PMF factor profiles (Figs. 2, 8, and 12), the species percentages vary from year to year. For example, in the case of the GEM re-emissions and biomass burning factor at SAT, the TGM percentage was 19%±10% (annual mean ± std. dev.). This temporal variability is captured by performing a separate PMF run for each year, but it would be missed if PMF was performed once for the entire time series. Source profiles can change over time, for example the fuel type and temperature of a forest fire can vary from one fire to another (and is influenced by climate variability); emissions control technology may impact the relative proportions of CO and TGM in the anthropogenic component of the Hg pool; Hg emissions from powerplants and industrial sources have decreased due to domestic regulations (Gov't of Canada, 2024; USEPA, 2024). Thus, a constant factor profile assumption over the long-term may not be valid in all cases. Performing a separate PMF run for each year requires more effort than a single PMF run for the entire time series, but it offers more insight into the consistency of the factor profiles year over year. The rationale behind performing separate yearly runs is discussed in the revised paper (section 2.2), which reads "Source profiles can change over time, for example the fuel type and temperature of a forest fire can vary from one fire to another (and is influenced by climate variability); emissions control technology may impact the relative proportions of CO and TGM in the anthropogenic component of the Hg pool; Hg emissions from powerplants and industrial sources have decreased due to domestic regulations. Thus, a constant factor profile assumption over the long-term may not be valid in all cases."

Government of Canada: National Pollutants Release Inventory: mercury, Canada's actions, https://www.canada.ca/en/environment-climate-change/services/national-pollutant-release-inventory/tools-resources-data/mercury.html, 2024.

USEPA: Mercury and Air Toxics Standards, https://www.epa.gov/stationary-sources-air-pollution/mercury-and-air-toxics-standards, 2024.

As suggested by the reviewer, we performed an additional PMF run for the entire time series. The results are discussed in the revised Supplement section S1, and the daily TGM source contributions are plotted in Fig. S11 (yearly runs are labelled as 'separate' and additional run for the full time

series is labelled 'all'). For the GEM re-emissions, biomass burning and Hg pool factors, the two runs produced similar seasonal patterns. However, there are noticeable differences for other sources. The separate runs show greater interannual variability in the daily TGM relative contributions compared to 'all' run scenario. For example, elevated relative contributions during 2010, 2011 and 2014 for local combustion. This is likely because the TGM percentages in each factor vary year to year as discussed above. We also see that the daily relative contributions from natural surfaces are elevated over the course of the time series for the 'all' run, whereas contributions from anthropogenic emissions were occasionally more important for the 'separate' runs. The 'separate' yearly runs capture the variability in the observations better than 'all' run. This is supported by the higher $R^2$ between modeled and observed TGM concentrations for the 'separate' case (0.75, Fig. S1) compared to 'all' case (0.68). The slope of the trendline for the 'separate' case is also closer to the 1:1 line (0.81 vs. 0.70).

The percentage contribution from natural surface emissions (wildfires plus re-emissions) from 'all' run was 70.6% on average, with a range of 66-76% depending on the year. This is higher compared with results from 'separate' runs; however, the conclusion that natural surface emission contribution exceeds anthropogenic contribution to TGM has not changed. Based on the 'all' scenario, the mean relative TGM source contributions were 50% from the Hg pool, 26.4% from terrestrial GEM re-emissions, 10.4% from shipping and SSA processing, 6.3% from oceanic evasion, 3.8% from wildfires, and 3.4% from local combustion. Most of the percentages were comparable to those obtained from 'separate' runs, except for higher terrestrial re-emissions and lower oceanic evasion. Secondary sulfate (regional Hg emissions and chemical transformation) did not contribute to TGM in the 'all' run case, whereas it contributed a few percent of the TGM in the 'separate' case.

Furthermore, I also do not understand why the authors did not merge all the sites into a single analysis. In a data-driven model like PMF, the number of receptors (sampling sites) is crucial for increasing the model's ability to discern distinct sources. By running the sites separately, the authors miss out on the advantage of having a broader spatial coverage and a larger dataset, both of which can significantly improve the resolution of source identification. Combining data from multiple receptors across different sites increases the model's power to detect and distinguish between sources, especially when there are overlapping emission signatures that might vary in strength and frequency across locations.

Response: The data size at each site and in each year is sufficient for PMF analysis. The three sites are very far apart from each other with the estimated distances (km) being: SAT-EGB (3290), SAT-KEJ (4355), and EGB-KEJ (1165) (see map below). The regional and local sources affecting the sites have potentially different emission signatures. For example, metal smelting and cement production are the primary industrial sources near SAT, while coal-fired power plants are near KEJ. Thus, it was preferred to derive the factor profiles unique to each site.

[Figure]

Increasing the number of receptors also enhances the model's capacity to address complex atmospheric dynamics, as it allows for a richer dataset that captures variations due to both local sources and regional transport processes. When multiple sites are analyzed together, the model has more information to work with, potentially identifying regional patterns that might be missed when each site is treated independently. The inclusion of multiple receptors also reduces the risk of overfitting the model to local conditions or short-term fluctuations at any single site, leading to a more generalizable and reliable source apportionment.

Response: As mentioned above, the data size at any single site is large enough. Each site has 8-14 years of 24-h data containing 12 variables. The long-term data coverage and high-temporal data resolution (Supplement Table S1) are sufficient for PMF modeling to capture variations in local and regional sources. This is clarified in section 2.2 of the revised paper. In this study, the PMF model captured both local source impacts (i.e., GEM re-emissions from land and nearby ocean, local combustion, local shipping, crustal/soil emissions) and regional source impacts (i.e., wildfires, regional combustion from U.S. northeast). The Hg pool was also inferred from the PMF model; this captures all types of Hg emissions mostly from the northern hemisphere and its impact on the receptor sites via long range transport.

It is not possible to include more receptor sites because some ancillary measurements are not available at other TGM sites. To clarify, a sentence was added in section 2.1, "The CAPMoN TGM sites were selected for this study to provide an update on current concentrations and patterns. The same set of ancillary measurements are also available at the sites to conduct PMF analysis." As mentioned above, the sites are very far apart; hence the data should not be combined considering the potentially different sources, meteorology, and transport patterns affecting the sites. There is minimal risk of the model overfitting to local conditions or short term variability because the sites are located in rural-remote areas infrequently impacted by large emission sources.

The number of species used in the factorization performed by the authors appears insufficient to properly apportion the sources they claim to resolve. The profiles of the factors presented do not

convincingly align with the expected "fingerprints" of the emission sources they attribute them to. For example, in Figure 12, the "local combustion" factor is primarily characterized by SO2 only, yet it lacks loading of CO, which is a well-known combustion tracer. Instead, CO is predominantly loaded in the "background" factor, which should typically be dominated by long-lived species. This misallocation raises concerns about the accuracy of the factor assignments and suggests that the model may not be adequately capturing the true source profiles.

Response: $SO_2$ is a short-lived pollutant (lifetime of a few days) and is emitted in large quantities from combustion sources, especially coal combustion and metal smelting which are also important anthropogenic Hg sources. This makes $SO_2$ a suitable local combustion tracer. In previous mercury source apportionment, $SO_2$ and Hg were typically used to assign profiles to local combustion (Eckley et al., 2013; Wang et al., 2013). While CO is also emitted from combustion sources, it has a relatively longer lifetime in the order of a few months, which allows for it to be transported over long distances (Jeffery et al., 2024). Given the relatively small magnitude of combustion sources near these rural and remote sites, it is not surprising that the CO emissions from these sources contribute a relatively small proportion of the total CO observed, which is dominated by the hemispheric background. This characteristic is similar to that of GEM and TGM. These are clarified in the revised paper (section 3.1.2). Instead of "background", we also renamed the factor to Hg pool, which encompasses all types of Hg emissions mostly from the northern hemisphere and its subsequent long distance transport. As shown in the global distribution of CO mixing ratios below, higher CO levels are broadly distributed across most of the northern hemisphere owing to fossil fuel combustion and biomass burning.

[Figure]

Paulo Penteado, NASA/JPL-Caltech, https://airs.jpl.nasa.gov/resources/239/airs-global-carbon-monoxide-over-20-years-2002-2022/

Eckley, C.S., Parsons, M.T., Mintz, R., Lapalme, M., Mazur, M., Tordon, R., Elleman, R., Graydon, J.A., Blanchard, P. and St. Louis, V., 2013. Impact of closing Canada's largest point-source of mercury emissions on local atmospheric mercury concentrations. Environmental Science and Technology, 47(18), 10339-10348.

Jeffery, P. S., Drummond, J. R., Zou, J., and Walker, K. A.: Identifying episodic carbon monoxide emission events in the MOPITT measurement dataset, Atmos. Chem. Phys., 24, 4253–4263, https://doi.org/10.5194/acp-24-4253-2024, 2024.

Wang, Y., Huang, J., Hopke, P. K., Rattigan, O. V., Chalupa, D. C., Utell, M. J., and Holsen, T. M., 2013. Effect of the shutdown of a large coal-fired power plant on ambient mercury species, Chemosphere, 92(4), 360-367.

Moreover, it seems that the authors constrained the number of factors to a level that exceeds what the available data can reliably cluster (with physical apportionment mean). The factorization results in some clusters that are difficult to justify from a source attribution perspective. For instance, there is a factor which unlikely represen a source, this suggests that the model may be overfitting, potentially driven by non-source-related variables.

There are also several indications in the presented profiles that attempting to resolve six sources from the limited number of species (or variables) used in the analysis constitutes an over-extrapolation of what is feasible through this factorization method. Some factors, for example, the cluster is basically loaded with Ca and Mg, other that is loaded with Cl and Na only. These elemental groupings suggest that the model is forming clusters based on chemical similarity only, rather than true source-specific emissions. This is further evidence that the factorization may be over-constrained, leading to artificial factors that do not accurately represent distinct sources.

Response: Twelve variables were included in the PMF model. The six model factors represent a variety of sources, which are all potential sources of Hg. The sources include not only well-established anthropogenic Hg sources (i.e. fossil fuel combustion), but also those that are less understood and less constrained in terms of the global Hg emissions budget, e.g. GEM re-emissions, wildfires, evasion of GEM from ocean, crustal/soil dust, and contributions from the Hg pool. Given that the sites are in rural-remote areas (infrequently impacted by large Hg point sources), the role of these less-studied sources are potentially important. Therefore, we used Cl and Na as tracers of a potential marine source of Hg, such as oceanic Hg evasion or shipping emissions (explained in section 3.1.2 of the revised paper). Ca and Mg are tracers of crustal/soil dust. In terrestrial ecosystems, Hg is derived either geologically or via atmospheric wet and dry (including litterfall) deposition, and is mainly bound to soil organic matter as oxidized Hg or Hg(II) (Eckley et al., 2016). Hg(II) also has a strong affinity for NaCl particles (Rutter and Schauer, 2007), which may explain the presence of TGM in the road salt factor. Hg can subsequently be re-emitted by wind erosion or land disturbance. This is explained in the revised paper (section 3.2.2).

The TGM percentage in the crustal/soil factor with high Ca and Mg and factors with high Cl and/or Na) are small; however, the values are not negligible (see Figures 2, 8 and 12). The annual TGM percentages are 8-10% on average and vary interannually. On a daily basis, TGM contributions can reach up to 90% for crustal/soil emissions and road salt (Fig. 10). In Figure 5, we can see another example where daily TGM contributions can reach up to 90% from shipping and SSA processing and oceanic evasion. The reviewer is concerned that these are artificial or non-significant sources of Hg; however, it is clear that their contributions to daily TGM can be important.

Eckley, C.S., Tate, M.T., Lin, C.J., Gustin, M., Dent, S., Eagles-Smith, C., Lutz, M.A., Wickland, K.P., Wang, B., Gray, J.E., Edwards, G.C., Krabbenhoft, D. P., and Smith, D. B.: Surface-air mercury fluxes across Western North America: A synthesis of spatial trends and controlling variables, Sci. Total Environ., 568, 651-665, 2016.

Rutter, A. P., and Schauer, J. J.: The effect of temperature on the gas–particle partitioning of reactive mercury in atmospheric aerosols. Atmos. Environ., 41(38), 8647-8657, 2007.

Based on the issues discussed, I recommend a major revision of the manuscript. It is clear that the authors have put significant effort into this work, but there are substantial improvements that need to be made, particularly in the source apportionment performing and analysis. I strongly encourage the authors to further explore the capabilities of source apportionment resource. While there is certainly a learning curve, catching up in this area will greatly enhance the robustness and accuracy of the study. I am confident that it will be worth the effort and will lead to more defensible results and insightful discussions.

Given the concerns raised about the current factorization and its interpretation, revisiting the apportionment process will likely result in significant changes to the manuscript's overall findings and discussion. I recommend that the authors reanalyze the data, addressing the over-extrapolation issues and ensuring that the profiles correspond more clearly to recognizable emission sources. Once the source apportionment is properly refined, I encourage the authors to resubmit the manuscript, as I believe it has the potential to make a valuable contribution to the field.

Response: The reviewer's concerns regarding the source apportionment analysis have been addressed in the responses and additional analysis and results have been incorporated in the revised paper. The major revisions to the manuscript are summarized below.

- Justification for including temperature in the PMF model (section 2.2)
- Clarification on the long term data coverage available at the sites (section 2.2)
- Justification for performing yearly model runs (section 2.2) and results of PMF model runs using the entire time series and comparison with the yearly runs (Supplement S1, Fig. S11)
- Discussion of PMF sensitivity runs using a different number of factors (Supplement S2, Tables S5-S7)
- Discussion of model fit and residuals plots for the final PMF solution (Supplement S3, Fig. S12)
- Renaming of background factor to Hg pool (throughout the manuscript)
- Clarification on using $SO_2$ as a tracer for local combustion and CO as a tracer of the Hg pool and long range transport (section 3.1.2)
- Intercomparison of model X and B TGM concentrations and related discussion (Supplement S4, Fig. S13)

In addition, I suggest that the authors expand on the differences between the two analyzers used (2537B and 2537X) as part of their revised manuscript. A thorough comparison of these instruments would be highly interesting, especially regarding any differences in performance or measurement outcomes. I would be very keen to see these comparisons and an error vector decomposition.

Response: Our assessment of the hourly TGM differences between Tekran 2537X and 2537B is published along with the quality controlled TGM dataset (ECCC, 2024). The monthly plots of TGM from model X and B are shown in Supplement Fig. S13. The model X and B analyzers at the EGB site operated side by side during Feb-Aug 2017. The model X reported slightly higher TGM than model B

with a mean hourly difference of 0.06 ng m$^{-3}$ (3.9%). Monthly mean hourly TGM differences were in the range of 0.02-0.1 ng m$^{-3}$ (1.4-6.3%). Model X and model B analyzers were also operated side by side at the same site during Mar-Jul 2018. The model X reported higher TGM than model B with a mean hourly difference of 0.08 ng m$^{-3}$ (6.4%). Monthly mean hourly TGM differences were in the range of 0.07-0.09 ng m$^{-3}$ (5.4-7.4%). Considering the differences were not significant and the concentrations showed similar trends, TGM concentrations from the model X and B analyzers were averaged during the periods when valid hourly data were available.

TGM was measured concurrently at KEJ and KEB from February to June in 2017. Note that KEJ and KEB sites are not co-located. KEJ was operating a model B analyzer; the site was relocated 3 km south of the original site in Feb 2017 (KEB) and the model X analyzer began operating at the new site. TGM was higher at KEB than at KEJ with a mean hourly difference of 0.26 ng m$^{-3}$ (18.8%). Monthly mean hourly TGM differences were 0.20-0.29 ng m$^{-3}$ (16.6-21.5%). These differences may be due to the different analyzer models and/or relocation of the monitoring site; the exact cause is inconclusive. Given the large TGM differences between KEB and KEJ, our decision was that the data from the two sites should not be combined into a single time series. Therefore, the 2017-2018 data at the new site were not used for PMF modeling and long-term trends analysis. We have added this discussion to Supplement section S4 and corresponding plots in Fig. S13.

Environment and Climate Change Canada: Canadian Air and Precipitation Monitoring Network (CAPMoN), Toronto, Ontario, Canada, Data files: AtmosphericGases-TGM-CAPMoN-AllSites-2017.csv, AtmosphericGases-TGM-CAPMoN-AllSites-2018.csv, https://doi.org/10.18164/e1df5764-1eec-4a9f-9c03-f515b396b717, 2024. **(Go to Downloads, views, and links > View ECCC Data Mart)**

---

## Author Response (AR2)

Dear Prof. Dommergue (ACP Editor):

We are submitting a second revision of our manuscript (EGUSPHERE-2024-2895), entitled "Natural Surface Emissions Dominate Anthropogenic Emissions Contributions to Total Gaseous Mercury at Canadian Rural Sites", for potential publication in Atmospheric Chemistry and Physics as part of the special issue, Mercury science to inform international policy: the Multi-Compartment Hg Modeling and Analysis Project (MCHgMAP) and other research.

We have addressed all the comments provided by Reviewer #2. Please see the enclosed authors' response to reviewers for details. The re-submission also includes a manuscript and supplement file with tracked changes. Thank you for taking care of the review process for this paper.

Sincerely,

Irene Cheng, Amanda Cole, Leiming Zhang, and Alexandra Steffen

Environment and Climate Change Canada (ECCC), Government of Canada

Correspondence to: irene.cheng@ec.gc.ca

**Response to Reviewer #2**

We appreciate the comments from this reviewer and have provided responses to all comments and improved on the manuscript.

Thank you to the authors for having addressed my comments. I have two comments.
First, Reviewer #1 made an excellent suggestion about pooling data from all years for the PMF analysis. Using data from all years not only increases statistical power but also makes sense in terms of interpreting the results. A constant factor profile enables comparability across years and allows for the identification of trends in the same set of factors throughout the study period at individual sites. Interannual variation in source composition is reflected in the time-dependent factor scores.

When running PMF annually, the top six factors were determined based on their contribution to the total variance of TGM in each year. In principle, interannual variations in the contribution of the top factors and in the annual total variance of TGM could compromise the comparability of the quantitative contributions of factors between years. However, in this case, the authors obtained the exact same top six factors every year despite running PMF separately for each year. This suggests that these six factors were consistently dominant throughout the entire study period. It is likely that their results would not differ significantly if they had pooled data from all years together.

Response: In response to the previous suggestion from Reviewer #1, we had added the results from pooling data from all years (single run) for the SAT site to Supplement section S1, which confirmed there was no significant change to the findings of the study. We have now carried out the single runs for the other two sites (EGB and KEJ). The single run factor profiles and TGM source contributions for all sites are now presented in the main paper. We have replaced the previous discussions, figures, and tables of the yearly runs in the main paper and supplement.

The model-observation agreement from the single run was equally as good as those of the yearly runs for the SAT and KEJ sites; however, it was not the case for the EGB site. TGM source contributions for EGB were compared between the single run and yearly runs, and the discussion was added in the main paper (section 4.1). The section reads:

**4.1 PMF yearly runs vs. single time series run**

Additional PMF runs were conducted separately for each year, and the TGM source contributions for the six factors were compared with those of the single time series run. While the results between the two sets of runs were comparable for SAT and KEJ, clear differences were found for EGB. The PMF modelled TGM concentrations derived from the yearly runs were better correlated with the observed TGM ($R^2$=0.71; Fig. S8) than that from the single run ($R^2$=0.43; Fig. S1). Furthermore, the yearly runs produced a better fit of the interannual variability and long-term trends in TGM. In the yearly runs, the mean relative contribution of anthropogenic emissions to annual TGM was 27.5% at EGB, which was greater than the single run scenario. The Hg pool (63%) contributed the most to annual TGM followed by terrestrial GEM re-emissions (15.4%), crustal/soil

dust (8.7%), local combustion (5.9%), road salt (4.1%), secondary sulfate (1.8%), and wildfires (1.3%), respectively. These percentages are lower for the Hg pool and GEM re-emissions and are higher for crustal/soil dust, local combustion, road salt, and secondary sulfate compared to those of the single run. In the yearly runs, TGM contributions from the Hg pool was the main driver of the observed TGM trend, whereas there was no dominant driver of the TGM trend from the single run.

Figure S9a illustrates the model-observation discrepancies occurred in 2005-2009 and 2012-2013 in the single run. During 2005-2009, TGM contributions from the Hg pool, local combustion, crustal/soil, and road salt for the single run were smaller compared to those of the yearly runs (Fig. S9b). In 2012-2013, TGM contributions from the Hg pool and GEM re-emissions for the single run were greater than those of the yearly runs. The differences between the single and yearly runs may be caused by various reasons. The underestimation of the Hg pool and local combustion contributions suggests a constant factor profile assumption in the single run may not be valid over the long term because of changes in the emissions control technology affecting the speciation profiles as discussed in Zhang et al. (2016). Perhaps the estimated TGM abundance in the Hg pool factor and local combustion factor should be higher in the earlier period. For crustal/soil, road salt and GEM re-emission contributions, dust emissions and GEM ($Hg^0$) flux are highly variable. These processes could depend on land disturbance from agricultural activities at EGB, road salt applied, and meteorology. For GEM flux, it is also affected by soil temperature and its Hg content, moisture level, organic matter content, vegetation and litterfall cover, Hg uptake, and ambient Hg concentration. The TGM content in dust is affected by gas-particle partitioning. These complex processes are perhaps not well captured in the factor profiles and contributions for terrestrial re-emissions in the single run but are better modelled in the yearly runs. It is recommended that short-term runs be performed across the time series to confirm the single run PMF results are robust.

Zhang, Y., Jacob, D.J., Horowitz, H.M., Chen, L., Amos, H.M., Krabbenhoft, D.P., Slemr, F., St. Louis, V.L. and Sunderland, E.M.: Observed decrease in atmospheric mercury explained by global decline in anthropogenic emissions, Proc. Natl. Acad. Sci., *113*(3), 526-531, 2016.

[Figure]

[Figure]

Figure S8: Comparison of PMF modelled and observed 24-h TGM (ng m$^{-3}$) for EGB using regression analysis (left) and time series analysis (right). PMF modelled TGM are based on yearly runs across the time series.

[Figure]

Figure S9: TGM annual source contributions (ng m$^{-3}$) at EGB for (a) single run and (b) yearly runs for 2005-2018. Bar graphs: PMF modelled TGM; red line: mean observed TGM.

I am unclear about what the authors mean by "variation" in their statement of "while the variability from year to year gives us an indication of the uncertainty in those profiles", and why they consider such variation to be "uncertainty in those profiles."

Response: Regarding the previous response to Reviewer #1, we agree that the statement was misleading. We are referring to the slight differences in the TGM concentrations in the factor profiles among the years when we performed the yearly runs. By contrast, the single run produces one profile which is constant across the time series. The factor profiles between the yearly runs and single run should be comparable if those six factors are prevalent throughout the time series as this reviewer noted, and this is a method that could be used to confirm the single run results are robust.

Second, the description of Figure 1 is inaccurate. What does "annual TGM" refer to? If it pertains to annual median values, then only EGB started to show a decreasing trend from 2009 to 2013 with the other two locations showing declines in later years. SAT started to see a return to higher values in 2014, while KEJ showed no such return, as its data ended in 2016. Additionally, it was unclear whether the decreasing trends before 2013 were statistically significant. Why were the values so different in 2006 and 2007 between EGB and KEJ?

Response: In the text, annual TGM refers to annual mean TGM. This has been clarified in sections 3.1.1, 3.2.1, and 3.3.1. The Fig. 1 caption now reads: "Box-whisker plots of annual summary statistics of 24-h average TGM concentrations (ng m$^{-3}$) at Egbert (EGB), Kejimkujik National Park (KEJ) and Saturna (SAT)". Based on the annual mean values, TGM decreased from 2009 to 2015 and then returned to higher concentrations in 2016 and 2018 at SAT. At EGB, the annual mean TGM decreased from 2005 to 2013 and then rebounded in 2014 with concentrations remaining stable during 2014-2018. At KEJ, the annual mean TGM decreased significantly from 2005 to 2006. A parabolic pattern was observed thereafter with concentrations increasing during 2006-2011 and then decreasing during 2012-2016.

Sections 3.1.1, 3.2.1, and 3.3.1 provide an overview of the TGM concentrations without examining the statistical significance. Note the long-term trends analysis (including the magnitude of the

trend and p-values showing the statistical significance level) is presented in sections 3.1.5, 3.2.5, and 3.3.5. It is clear from Fig. 6 (blue line graph) there was a decreasing trend in the observed TGM between 2006 and 2013 for EGB.

The large TGM difference between EGB and KEJ in 2006-2007 can be explained by the Hg emission patterns. Around EGB, Hg emissions based on the Canadian National Pollutant Release Inventory were high during 2006-2007 and only decreased sharply after 2008 followed by an increase starting in 2010 (Fig. S4). Around KEJ, Hg emissions decreased sharply in 2006 and then again in 2007 (Fig. S7), which was due to emissions reductions and the closure of a power plant.

[Figure]

Figure 6: Long-term trends in observed and PMF modeled TGM concentrations at SAT, EGB and KEJ. Blue line: observed or modeled TGM; red line: trendline; green text: slope of the trendline (ng m$^{-3}$ yr$^{-1}$)

[Figure]

Figure S4: Hg emission sources within 150 km of EGB in Province of Ontario (ECCC NPRI, 2023)

[Figure]

Figure S7: Hg emission sources within 150 km of KEJ in Provinces of Nova Scotia and New Brunswick (ECCC NPRI, 2023)